# Monsoonal forcing of cold-water coral growth off south-eastern Brazil during the past 160 kyrs

André Bahr[1], Monika Doubrawa[1,2], Jürgen Titschack[3,4], Gregor Austermann[1], Andreas Koutsodendris[1], Dirk Nürnberg[5], Ana Luiza Albuquerque[6], Oliver Friedrich[1], Jacek Raddatz[7]

[1]Institute of Earth Sciences, Heidelberg University, Im Neuenheimer Feld 234, 69120 Heidelberg, Germany
[2]Earth and Environmental Sciences, KU Leuven, Celestijnenlaan 200e, 3001 Leuven, Belgium
[3]MARUM – Center for Marine Environmental Sciences, University of Bremen, Leobener Str. 8, 28359 Bremen, Germany
[4]Senckenberg am Meer, Marine Research Department, 26382 Wilhelmshaven, Germany
[5]GEOMAR Helmholtz Centre for Ocean Research, Wischhofstraße 1-3, 24148 Kiel, Germany
[6]Departamento de Geoquímica, Universidade Federal Fluminense, Outeiro São João Baptista s/n. – Centro, Niterói, RJ, Brazil
[7]Institute of Geosciences, Goethe University, Frankfurt, Altenhöferallee 1, 60438 Frankfurt am Main, Germany

*Correspondence to*: André Bahr (andre.bahr@geow.uni-heidelberg.de)

**Abstract.** Cold-water corals (CWC) constitute important deep-water ecosystems that are under increasing environmental pressure due to ocean acidification and global warming. The sensitivity of these deep-water ecosystems to environmental change is demonstrated by abundant paleo-records drilled through CWC mounds that reveal characteristic alterations between rapid formation and dormant or erosive phases. Previous studies have identified several central parameters for driving or inhibiting CWC growth such as food supply, oxygenation, and the carbon saturation state of bottom water, yet there are still large uncertainties about the relative importance of the different environmental parameters. To advance this debate we have performed a multi-proxy study on a sediment core retrieved from the 25 m high Bowie Mound, located at 866 m water depth on the continental slope off south-eastern Brazil, a structure built up mainly by the CWC *Solenosmilia variabilis.* Our results indicate a multi-factorial control on CWC growth at Bowie Mound during the past ~160 kyrs, which reveals distinct formation pulses during northern high latitude glacial cold events (Heinrich Stadials, HS) largely associated with anomalously strong monsoonal rainfall over the continent. The ensuing enhanced run-off elevated the terrigenous nutrient and organic matter supply to the continental margin, and likely have boosted marine productivity. The dispersal of food particles towards the CWC colonies during HS was facilitated by the highly dynamic hydraulic conditions along the continental slope that prevailed throughout glacial periods. These conditions caused the emplacement of a pronounced nepheloid layer above Bowie Mound thereby aiding the concentration and along-slope dispersal of organic matter. Our study thus emphasizes the impact of continental climate variability on a highly vulnerable deep-marine ecosystem.

## 1 Introduction

Cold-water corals (CWC) are hotspots of biodiversity in the deep-sea (Roberts and Cairns, 2014), important constituents of the deep water carbon cycle (Lindberg and Mienert, 2005; Titschack et al., 2009; White et al., 2012; Cathalot et al., 2015; Titschack et al., 2015, 2016), and potent bio-engineers due to their sediment baffling capacity that allows for enormous sediment accumulation rates of up to 1500 cm/kyr during maximum CWC mound formation phases (Titschack et al., 2015; Wienberg and Titschack, 2017, Wienberg et al., 2018). Yet the impact of global climate change on CWC reefs and the associated ecosystems under is poorly constrained, because the factors driving or inhibiting their occurrence and the potential thresholds in their resilience to environmental change are still under debate (Hebbeln et al., 2019; Raddatz and Rüggeberg, 2019). Geological records reveal that coral mounds typically exhibit distinct phases of formation, often intercalated by intermittent periods of non-deposition and/or potentially erosion, indicating a high sensitivity of CWCs to changing boundary conditions (e.g., Rüggeberg et al., 2005; Kano et al., 2007; Frank et al., 2011; Raddatz et al., 2014, 2016; Wienberg and Titschack, 2017; Wienberg et al., 2018).

The most common framework-forming CWC comprise of *Lophelia pertusa* (recently assigned to the genus *Desmophyllum* by Addamo et al., 2016), *Macropora oculata*, *Solenosmilia variabilis*, *Bathelia candida*, and *Enallopsammia profunda* (e.g., Mangini et al., 2010; Frank et al., 2011; Muñoz et al., 2012; Hebbeln et al., 2014; Raddatz et al., 2020). Field and laboratory studies of *L. pertusa*, the most intensively investigated species, suggest that scleractinian CWC are non-specialists regarding food sources, which range from particulate to dissolved organic carbon (POC and DOC, respectively) (Kiriakoulakis et al., 2005; Duineveld et al., 2007; Gori et al., 2014; van Oevelen et al., 2016), algae, bacteria, and zooplankton (Gori et al., 2014; Mueller et al., 2014). However, we note that similar studies on the feeding preferences of *S. variabilis*, the dominant framework-building CWC at the herein investigated Bowie Mound (Raddatz et al. 2020) are still missing. Additionally, changes in the properties and spatial configuration of ambient intermediate- or deep-water masses may also strongly impact CWC through changes in the dissolved oxygen concentration and the seawater parameters pH, alkalinity and carbonate-ion concentration. All these parameters affect the capacity of CWC to build their aragonitic framework (e.g., Form and Riebesell, 2012; Maier et al., 2012; Lunden et al., 2014; Hennige et al., 2015; Büscher et al., 2017; Auscavitch et al. 2020). Spatial fluctuations of intermediate- to deep-water masses further influence the depth and strength of pycnoclines, which are thought to play an important role in the concentration and dispersal of nutrients and food utilized by CWC (Frederiksen et al., 1992; Duineveld et al., 2007; Mienis et al., 2007; Rüggeberg et al., 2016). Aside from processes directly affecting the water-mass properties bathing CWC, several studies also point to the importance of sea surface productivity in providing food to the deep ocean (Davies et al., 2009; Soetaert et al., 2016). However, despite the proximity of CWC mounds situated on the continental slope to adjacent continents, the role of terrestrial nutrient and POC input is still a matter of debate (Wienberg et al., 2010; Hanz et al., 2019; Fentimen et al., 2020).

To systematically test the relative importance of the diverse factors potentially influencing CWC growth and mound formation, we investigated the response of CWC at Bowie Mound, a coral-bearing mound in the Campos Basin on the continental slope offshore south-eastern Brazil (Bahr et al., 2016; Raddatz et al., 2020) to changes in paleoenvironmental conditions. The presence of CWC-bearing mounds off Brazil was first reported by Viana et al. (1998) and Sumida et al. (2004) along the continental slope at intermediate water depths between 500 to 1000 m, bathed by Antarctic Intermediate Water (AAIW). At Bowie Mound, the dominating species is *S. variabilis*, which is

adapted to colder (as low as 3-4°C; Fallon et al., 2014; Flögel et al., 2014; Gammon et al., 2018) and less aragonite
saturated waters than *L. pertusa* (Thresher et al., 2011; Flögel et al., 2014; Bostock et al., 2015; Gammon et al., 2018).
The selected location at Bowie Mound is ideally suited to assess a variety of external factors capable of driving CWC
growth dynamics as it is situated at the interface of distinctly different water masses (cf. Section 2) and is strongly
influenced by terrigenous input from land and the broad shelf off Cabo Frio, which experiences intense seasonal
upwelling. This setting allows us to test the relative importance of different factors that are potentially crucial for the
CWC growth at Bowie Mound, in particular (i) intermediate water-mass variability via its impact on nutrient (e.g., Fe,
P, N) concentration, food availability, and local hydrodynamics and (ii) variations in nutrient and organic matter fluxes
derived from upwelling and terrestrial input in the context of global climatic changes. Our unique set of multi-proxy
data combined with Th/U-dated CWC demonstrates that an invigorated continental hydroclimate played a thus far
underestimated role in triggering CWC growth at the south-eastern Brazilian margin – a scenario that is likely affecting
CWC-mounds worldwide.

**2 Hydrological, climatological and geological setting**

The (sub)surface circulation in the western Tropical South Atlantic at the Campos Basin off southeast Brazil is
dominated by the southward flowing, warm Brazil Current (BC; Fig. 1). The BC forms the western portion of the
anticyclonic subtropical gyre (Stramma and England, 1999), which is characterized by high evaporation rates that
produced the Salinity Maximum Water (SMW, 24ºC, $\sigma_\theta \sim 25.2$) in the upper 200 m of the water column. The interaction
of the BC with the coastal hydrographic system promotes subsurface upwelling of South Atlantic Central Water
(SACW) on the shelf edge and on the shelf (Roughan and Middleton, 2002; Aguiar et al., 2014). Upwelling is
particularly strong during austral spring and summer when northeasterly winds generating upward Ekman pumping on
the mid shelf (Castelao and Barth, 2006; Castelao, 2012), which fuels productivity due to the subsurface encroachment
of nutrient-rich SACW. The SACW is found from below the SMW up to 500 m water depth and is characterized by
decreasing temperatures and salinities (20ºC, 36.0 psu to 5ºC, 34.3 psu; Fig. 1) (Raddatz et al., 2020) owing to its
formation in the southwest Atlantic and the South Indian Ocean (Sverdrup et al., 1942; Stramma and England, 1999).
The AAIW (34.3 PSU, ~4°C; Fig. 1) lies below the SACT and above North Atlantic Deep Water (NADW) which is
present below 1100 m water depth and has higher oxygen concentrations and salinities compared to AAIW (Mémery
et al., 2000). Below ~2500 m the Antarctic Bottom Water (AABW) constitutes the deepest and most dense water mass
in this region (Stramma and England, 1999).
The interaction between the north- or southward-directed flow of the different water masses with the morphology of
the slope at Campos Basin froms strong geostrophic currents (Viana and Faugères, 1998; Viana et al., 1998; Viana,
2001). These currents are responsible for enhanced sediment focusing leading to the formation of drift bodies, while
internal waves at the boundary between different water masses create wide-spread erosional surfaces (Viana et al.,
1998, 2001).
Bowie Mound itself has a total elevation of 25 m and is situated within a field of mound-like structures, which are
presently barren of living framework-forming CWC (Bahr et al., 2016). Located at 866 m water depth, it lies within
the core of the AAIW. Hence, it can be expected that changes in the nutrient and organic matter inventory of the AAIW

(Poggemann et al., 2017) and/or displacement of the intermediate water mass may have had a direct impact on the hydrodynamic conditions at Bowie Mound.

The Campos Basin receives freshwater and sediment input primarily from the Paraíba do Sul River, which delivers between 180 to 4400 m$^3$ s$^{-1}$ water (Carvalho et al., 2002) and 30 t yr$^{-1}$ of sediment (Jennerjahn et al., 2010) to the study area. Most precipitation in the hinterland of the Paraíba do Sul River occurs during austral summer, when strong

atmospheric convection forms the South Atlantic Convergence Zone, an elongated band of heavy precipitation that reaches from central Amazonia into the tropical South Atlantic (Carvalho et al., 2004; Marengo et al., 2012) (Fig. 1).

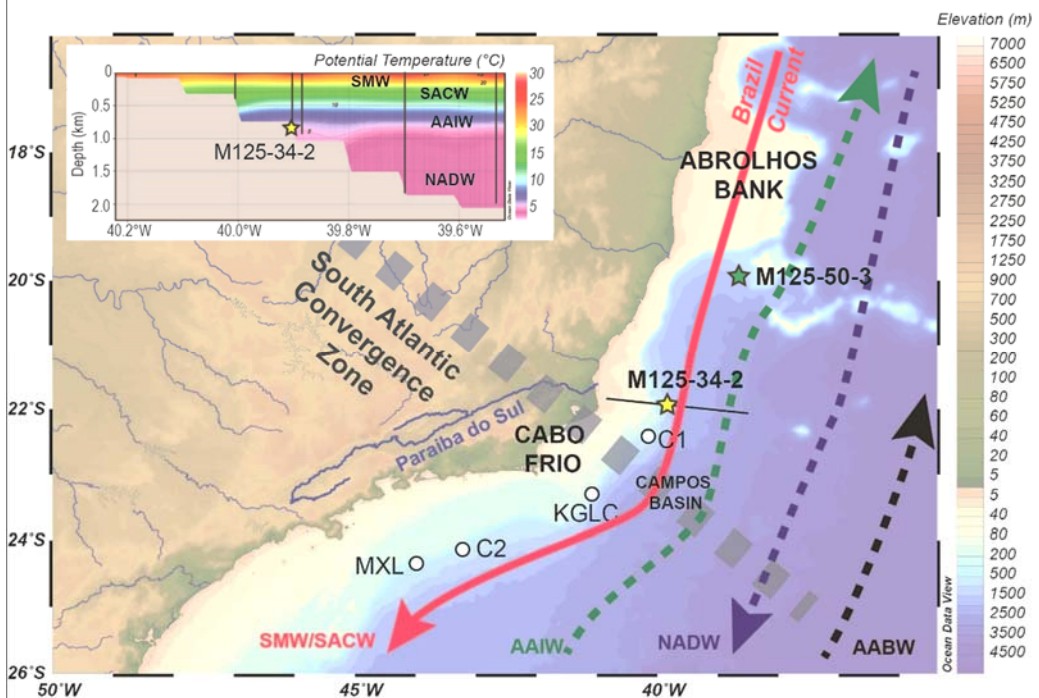

**Fig. 1** Location of core M125-34-2 (yellow star) on Bowie Mound, reference core M125-50-3 (green star) and other
CWC records published in Mangini et al. (2010) and Ruckelshausen (2013) (white dots). Major surface (red line), intermediate (green) and deep-water circulation features (blue) and water masses are indicated as well as the approximate location of the South Atlantic Convergence Zone (stippled line) as the main atmospheric feature. Inset shows a hydrographic section crossing the location of Bowie Mound core M125-34-2 with potential temperatures as measured via CTD (black lines) during R/V METEOR Expedition M125 (Bahr et al., 2016; Raddatz et al., 2020); the
location of the hydrographic section is indicated by a black line in the map. Figure modified after Raddatz et al. (2020). AABW – Antarctic Bottom Water, AAIW – Antarctic Intermediate Water, NADW – North Atlantic Deep Water, SACW – South Atlantic Central Water, SMW – Salinity Maximum Water.

## 3 Material and Methods

### 3.1 Material

Gravity core M125-34-2 was retrieved during R/V METEOR cruise M125 from the top of the 25 m high Bowie Mound in 866 m water depth at 21°56.957'S and 39°53.117'W (exact positioning was secured by the Ultra-Short-Baseline

system POSIDONIA; Fig. 1; Bahr et al., 2016). The core was cut into 1 m segments onboard and stored unopened at -20°C. After CT-scanning, the core was opened in the frozen state (for details on the CT-scanning cf. Skornitzke et al., 2019; Raddatz et al., 2020). Discrete samples for X-ray diffractometry (XRD), grain-size analyses, C and N content, and stable carbon and oxygen isotope analyses of foraminifera and organic matter were taken from the sediment matrix avoiding the sampling of coral fragments. X-ray fluorescence (XRF) scanning was performed on the archive halves, while avoiding coral segments when defining the sampling path of the detector.

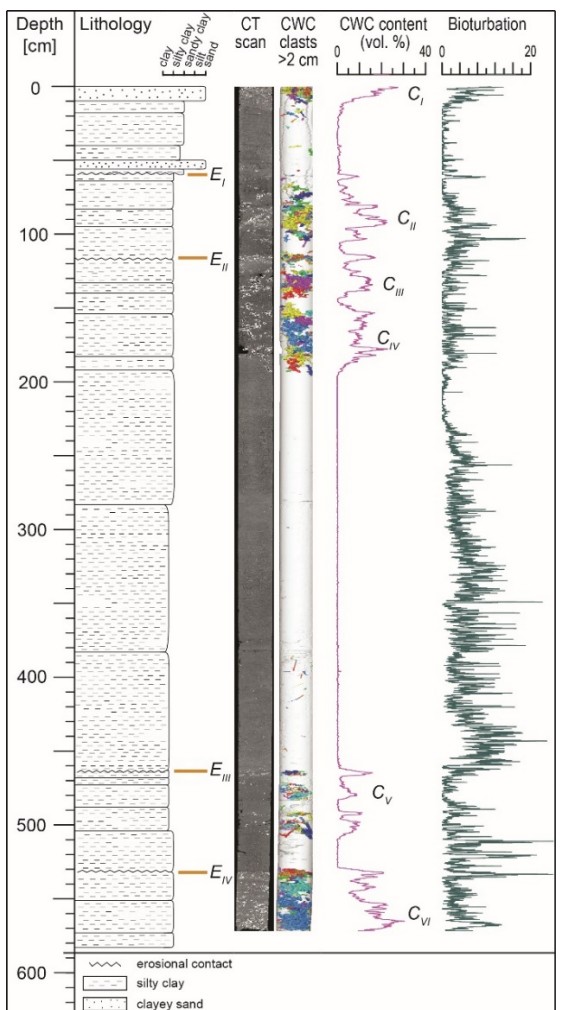

**Fig. 2.** Lithological log of Core M125-34-2 (21°56.957'S, 39°53.117'W, 866 m water depth) including CT-scanning image and CWC clasts >2 cm, CWC content and bioturbation index. Erosional surfaces $E_I$–$E_{IV}$ are indicated as well as CWC-bearing intervals $C_I$–$C_{VI}$ after Raddatz et al. (2020).

Core M125-34-2 (Fig. 2) consists of moderately to strongly bioturbated olive grey to dark grey, silty clayey sand with alternating coral-bearing and coral-free zones (Raddatz et al., 2020). The sediments are rich in micro- and macrofossils including pteropods, bivalves as well as benthic and planktonic foraminifers. Six intervals ($C_I$–$C_{VI}$) of particularly high coral contents of up to 31 vol.% were identified within the core at 0–13 cm, 80–105 cm, 131.5–138 cm, 157–190 cm, 479.5–480.5 cm, and 549.5–568 cm (Fig. 2; Raddatz et al., 2020). Coral-bearing intervals are characterized by the presence of *S. variabilis* and to a minor degree *M. oculata* as the major macrofossils (clast length >2 cm). Visual inspection and CT imaging (Appendix Figures A-F) reveal four prominent erosive surfaces ($E_I$–$E_{IV}$) with abundant coral fragments and shell debris at 58 cm, 117 cm, 465 cm, and 532 cm (Fig. 2).

To constrain the chronostratigraphy of core M125-34-2 and to assess the intermediate-water variability, adjacent core M125-50-3 (19°56.957'S; 38°35.979'W; 904 m water depth; Fig. 1) was sampled at 10 cm intervals for $\delta^{18}O$ analyses. Core M125-50-3 is barren of CWC and consists of bioturbated, greenish-grey hemipelagic mud with darker and lighter intervals. Two sandy, foraminifera-rich layers at 1226 and 1230 cm might point to periods of condensed sedimentation, otherwise no unconformities or erosive surfaces could be identified.

**3.2 Quantitative analysis of CT-scanning data**

Prior to opening, the sediment core sections were scanned with a SOMATOM Definition Flash computer tomograph at the Clinic of Diagnostic and Interventional Radiology (DIR) of Heidelberg University Hospital, Heidelberg, Germany, with 140 kVp tube potential and 570 mAs tube current – time product with a pitch of 0.4 (for details see Skornitzke et al., 2019). The raw data with a resolution of 0.5 mm in z-orientation and 0.3 mm in xy-orientation was reconstructed iteratively (ADMIRE, Siemens Healthineers) using a sharp kernel (I70 h level 3) to an isotropic voxel size of 0.35 mm. Further data processing was carried out with the ZIB edition of the Amira software (Stalling et al. 2005; http://amira.zib.de). Within Amira, the core sections were virtually reunited and the core liner and marginal coring artefacts were removed (~2 mm of the core rim). Furthermore, coral clasts were segmented and separated with the ContourTreeSegmentation module (threshold: 1400; persistence value: 1150) and quantified. Macrofossils >2 cm were visualized as surfaces in 3D following the methodology of Titschack et al. (2015). As an index for bioturbation, we determined the standard deviation of the matrix sediment X-Ray attenuation within each XY-oriented CT slice. The matrix sediment was segmented by selecting the data volume surrounding the corals, removing areas with values <500 HU (considered to represent air and water) and reducing the remaining segmented volume by three voxels to avoid marginal artefacts in the X-Ray attenuation caused for example by the resolution depending averaging effect.

**3.3 X-Ray fluorescence (XRF) scanning**

XRF core scanning was performed on the archive halves of the split core. All segments of core M125-34-2 were scanned at Heidelberg University, using an Avaatech (GEN4) XRF Core Scanner. XRF core scanner data were collected every 1.0 cm down-core with a 1.2 cm cross-core slit size. The split core surface was cleaned and covered with a 4 μm thin SPEXCerti Prep Ultralene1 foil to avoid contamination of the XRF measurement unit. Core intervals with very abundant corals were skipped to avoid damage of the foil covering the detector through sharp and rigid edges of coral fragments. Data was collected in two separate runs using generator settings of 10 and 30 kV and currents of 0.2 and 1.0 mA, respectively. Sampling time was set to 20 seconds per measurement. To counteract artifacts derived from variations in sediment porosity, water content and surface roughness, element counts were normalized by dividing the value of the component (C) by the sum of the counts for each depth (Bahr et al., 2014).

**3.4 X-Ray diffractometry (XRD)**

About 9 g of wet material was dried in an oven at 40°C and subsequently milled in a ball mill (Pulverisette, FRITSCH), with 300 cycles per sec for 3 minutes to obtain a powder with a grain size of 1-3 μm. The XRD measurements were carried out at the Institute of Earth Sciences, Heidelberg University, using a Bruker, D8 ADVANCE Eco diffractometer

(40 kV, 25mA) with a Cu Kα diode. The samples were measured in rotating, circular, synthetic sample holders. An angular range of 2θ from 5° to 70° was measured with a step size of 0.02 increments (3338 steps per sample) for 1 sec per step. Peak positions and intensity of data were analyzed with Diffract Suite EVA (Bruker Software). The Rietveld refinement program DIFFRAC.TOPAS (Bruker Software) was used to perform quantitative phase analysis.

**3.5 Grain-size analysis (Sortable Silt, $\overline{SS}$)**

The preparation of the samples for the $\overline{SS}$ analyses followed Bianchi et al. (2001) and Stuut et al. (2002). Wet samples with a weight between 0.3 and 1 g were dried over night at 40 °C. Macroscopically visible coral fragments were removed. After weighting the dry samples, 10 ml of 30 % $H_2O_2$ were added to each sample to dissolve the organic material under sub-boiling conditions on a heating plate until the reaction ceased. To remove carbonates, 5 ml HCl (10
%) were added to each sample under sub-boiling conditions for at least 10 minutes until the end of the reaction. Samples were washed through standard sieves (63 µm mesh size) to remove the sand and coarser fraction. The fraction >63 µm was dried at 40°C and weighed. The material <63 µm was transferred into 1 L beakers and filled to the top with demineralized water. After a settling period of at least 8 hours (h), the supernatant water was decanted, and the sample transferred into a 200 mL beaker, topped with demineralized water and settled for 8 h. After decanting supernatant
water, the sampled were transferred into 50 mL plastic beakers and put into an ultrasonic bath to disintegrate aggregated sediment particles. Afterwards, 35 ml Na-Pyrophosphate was added to prevent particles from forming new aggregates and 2 ml isopropyl alcohol minimize the formation of air bubbles in the liquid. Samples were measured with a Laser Particle Sizer (LPS) ANALYSETTE 22 by FRITSCH™ at the Institute of Earth Sciences, Heidelberg University, in wet dispersion covering the range 0.08-2000 µm with 99 size classes. The raw data was analyzed with the software
MaScontrol (FRITSCH™) by applying a Fraunhofer model. Results are derived from a total of five analytical runs with 100 scans per sample. Up to seven replicate samples were performed for each depth to minimize the natural variability of the samples. While there is relatively large inter-sample variability that is likely caused by the high amount of mica flakes present in the sediment distorting the laser beam, we nevertheless consider the results retrieved via the ANALYSETTE 22 as reliable (for an in-depth discussion see Jonkers et al., 2009).

**3.6 Carbon and nitrogen content**

Sediment samples were homogenized with a mortar and pestle and then weighted (0.5 mg for each analyses). The total organic carbon (TOC) and inorganic carbon (TIC, together TC) content was analyzed with a LECO RC-412 (Institute of Geoscience, Goethe University Frankfurt). The reproducibility of the replicate analyses was < 0.1 %. The total
nitrogen (TN) content was analysed with a LECO TruSpec Macro (Institute of Geoscience, Goethe University Frankfurt).

**3.7 Stable oxygen and carbon analysis**

**3.7.1 Stable isotope analysis of foraminiferal calcite**

For stable isotope analyses of core M125-34-2, samples were taken at 5–10 cm intervals, wet-sieved (>63 µm) and then dried. For each sample 1–3 tests of *Uvigerina* spp. (*U. peregrina* and *U. proboscidea*) or *Planulina wuellerstorfi*

were selected from the size fraction >125 µm, depending on availability. Stable carbon and oxygen isotope ratios were measured on a Thermo Fischer MAT 253 Plus IRMS gas isotope ratio mass spectrometer with coupled Kiel IV automated carbonate preparation device at the Institute of Earth Sciences, Heidelberg University. The instrument was calibrated using the in-house standard (Solnhofen limestone), which is calibrated against the IAEA-603. Values are reported versus the VPDB (Vienna Peedee Belemnite) standard. Standard deviations derived from repeated measurements of the internal standard are ±0.06 ‰ for stable oxygen isotopes ($\delta^{18}O$) and ±0.03 % for stable carbon isotopes ($\delta^{13}C$).

The ~13 m long core M125-50-3 was sampled every ~10 cm for benthic foraminiferal isotope analysis. The samples were freeze-dried and washed through a >63 µm sieve to separate the coarse and fine fractions. Specimens of the benthic genera *Uvigerina* spp. were hand-picked from the size fraction 315–400 µm. $\delta^{18}O$ analyses were performed on a ThermoScientific MAT 253 mass spectrometer with an automated Kiel IV Carbonate Preparation Device at GEOMAR. The isotope values are calibrated versus the NBS19 (National Bureau of Standards) carbonate standard and the in-house "Standard Bremen" (Solnhofen limestone). Isotope values presented in the delta-notation are reported in permil (‰) relative to the VPDB scale. The analytic error is ±0.06 ‰ for $\delta^{18}O$ and ±0.03 ‰ for $\delta^{13}C$.

**3.7.2 Stable isotope analysis of organic matter**

Previously homogenized samples were decarbonized with 10 % HCl to remove all inorganic carbon. Afterwards, the samples were centrifuged and washed several times with deionized water in order to remove residual HCl. The samples were then dried in an oven at 50 °C. Subsequent analyses of the carbon isotopic composition of organic carbon ($\delta^{13}C_{org}$) was performed by a Flash Elemental Analyzer 1112, connected to the continuous flow inlet system of a MAT 253 gas source mass spectrometer (Institute of Geosciences, Goethe University Frankfurt). Samples and standards both reproduced within ±0.2‰ and are reported relative to the VPDB standard.

**3.7 Statistical analysis**

Correlation coefficients and associated p-values were calculated using the Monte-Carlo-based SurrogateCorr function implemented in the "astrochron" package in R (Meyers, 2014) with 1000 iterations. This method has been particularly developed to assess the correlation of parameters sampled on different down-core resolutions (e.g. XRF scanning data *vs*. discrete grain size measurements; Meyers, 2014).

Discriminant analysis was performed with the program PAST (v3.15) (Hammer et al., 2001). Prior to analysis, the data was detrended and normalized to the mean and standard deviation.

**4. Age model refinement**

The age model of core M125-34-2 used in this study represents a refined version of the stratigraphy published in Raddatz et al. (2020), which is based on $^{230}$Th/U dates of CWCs. The six coral-bearing intervals, as described in Section 3.1, exhibited a mean accumulation rate of 30 cm kyr$^{-1}$ with an overall range from 2 cm kyr$^{-1}$ ($C_V$) to 80 cm kyr$^{-1}$ ($C_{III}$) (Raddatz et al., 2020).

To better constrain the age of CWC-barren intervals and evaluate the chronostratigraphic duration of potential hiatuses, we compared the $\delta^{18}O$ record of core M125-34-2 with the benthic isotope record of adjacent core M125-50-3. The clear glacial-interglacial pattern of $\delta^{18}O_{Uvi}$ in core M125-50-3 allows for the construction of a robust age model for this site by tuning its benthic isotope record to the LR04 benthic stack (Lisiecki and Raymo, 2005) and indicates that it extends to ~135 ka (Fig. 3). Its stratigraphic range is thus only slightly shorter than the one of M125-34-2 (158 ka based on Th/U dates) and is therefore suitable as an off-mound reference site. As both cores are situated in similar water depths and are thus bathed by the same water mass (today the AAIW, see Section 2), we expect the respective $\delta^{18}O$ values to not only follow a common glacial/interglacial pattern but also to be comparable in their absolute values, allowing for further constraining the chronostratigraphy of M125-34-2.

As neither shallow infaunal *Uvigerina* spp. and epibenthic *P. wuellerstorfi* were consistently present throughout core M125-34-2, we generated a spliced record of both species. For the $\delta^{18}O$ splice we corrected values of *P. wuellerstorfi* by adding the correction factor of +0.47 ‰ according to Marchitto et al. (2014). The resulting combined $\delta^{18}O$ record of M125-34-2 exhibits a considerable scatter from the top to erosive horizon $E_I$ including relatively depleted $\delta^{18}O_{Plan}$ values as low as 2.5 ‰ (Fig. 3), which puts this interval into a transitional phase between glacial and interglacial $\delta^{18}O$ levels. Neither $\delta^{18}O$ values nor absolute dating supports the preservation of Holocene deposits at the top of the gravity core. A deglacial age for the deposition of this interval is further corroborated by Th/U dates of 13.7 to 14.3 ka, respectively. Samples from between $E_I$ and $E_{II}$ show slightly heavier $\delta^{18}O$ values and Th/U dates clustering around 16.5 ka, which indicates a post-LGM deposition. The existing Th/U dates suggest that the hiatus represented by the erosive unconformity $E_I$ is most likely shorter than 2 kyr. The section between $E_{II}$ and $E_{III}$ on the other hand has relatively uniform $\delta^{18}O$ values around 4.3 ‰ ($\delta^{18}O_{Uvi}$) and 3.4 ‰ ($\delta^{18}O_{Plan}$), respectively, which matches M125-50-3 $\delta^{18}O_{Uvi}$ values during MIS 4 to 2. A Th/U date at the top of this section at 117 cm reveals an age of 34 ka, while CWC ages from slightly deeper in the core (between 131-190 cm) fall within the range of 60–63 ka (MIS 4). As $\delta^{18}O_{Uvi}$ values between $E_{II}$ and $E_{III}$ are less depleted than MIS 5 samples of reference site M125-50-3, we infer that those sediments were most likely deposited during MIS 4 and did not reach into MIS 5. Hence, it hence appears that deposits of MIS 2 and large parts of MIS 3 are not present in core M125-34-2, either due to non-deposition or subsequent erosion (note the prominent erosive surface $E_{II}$). This age assignment would also imply that the extended CWC-free portion from 200 to 465 cm was deposited within a short period of approximately 8 kyr during MIS 4 (62.2 ka as the oldest Th/U dates and ~70 ka as the MIS 4/5 boundary). This would yield a sedimentation rate of 33 cm kyr$^{-1}$ (Fig. 3A), which is typical for contouritic sediments at the south-eastern Brazilian margin (Viana et al., 1998; Hernández-Molina et al., 2014; Rebesco et al., 2014). The following interval between $E_{III}$ and $E_{IV}$ has relatively depleted $\delta^{18}O_{Uvi}$ values that fall into the range of MIS 5a–d values in reference core M125-50-3. Th/U dates of ~107 ka for CWC below $E_{III}$ support a deposition during MIS 5d and further indicate a prolonged interval of non-deposition or erosion leading to the absence of MIS 5a-c in core M125-34-2. As $\delta^{18}O$ values below $E_{IV}$ are again on glacial levels, this section can be assigned to MIS 6 which is in line with Th/U dates of 152.6–158.4 ka. Hence the penultimate interglacial MIS 5e is likely not recovered and falls into the hiatus represented by $E_{IV}$.

In summary, deposition at Bowie Mound site M125-34-2 appears to be concentrated during glacial intervals of intermediate ice volume and reduced during interglacial periods. The erosive horizons present in core M125-34-2 caused by winnowing due to strong current activity and internal waves provide evidence for extreme variability in the

hydrological regime at the south-eastern Brazilian Margin (Viana and Faugères, 1998; Viana et al., 1998; Viana, 2001). While winnowing has the capacity to remove the sediment matrix leading to lag deposits, it is rather unlikely that it would also remove coral fragments. Hence, it is feasible to assume that the dated intervals of CWC presence represent the complete sequence of CWC presence at this part of Bowie Mound.

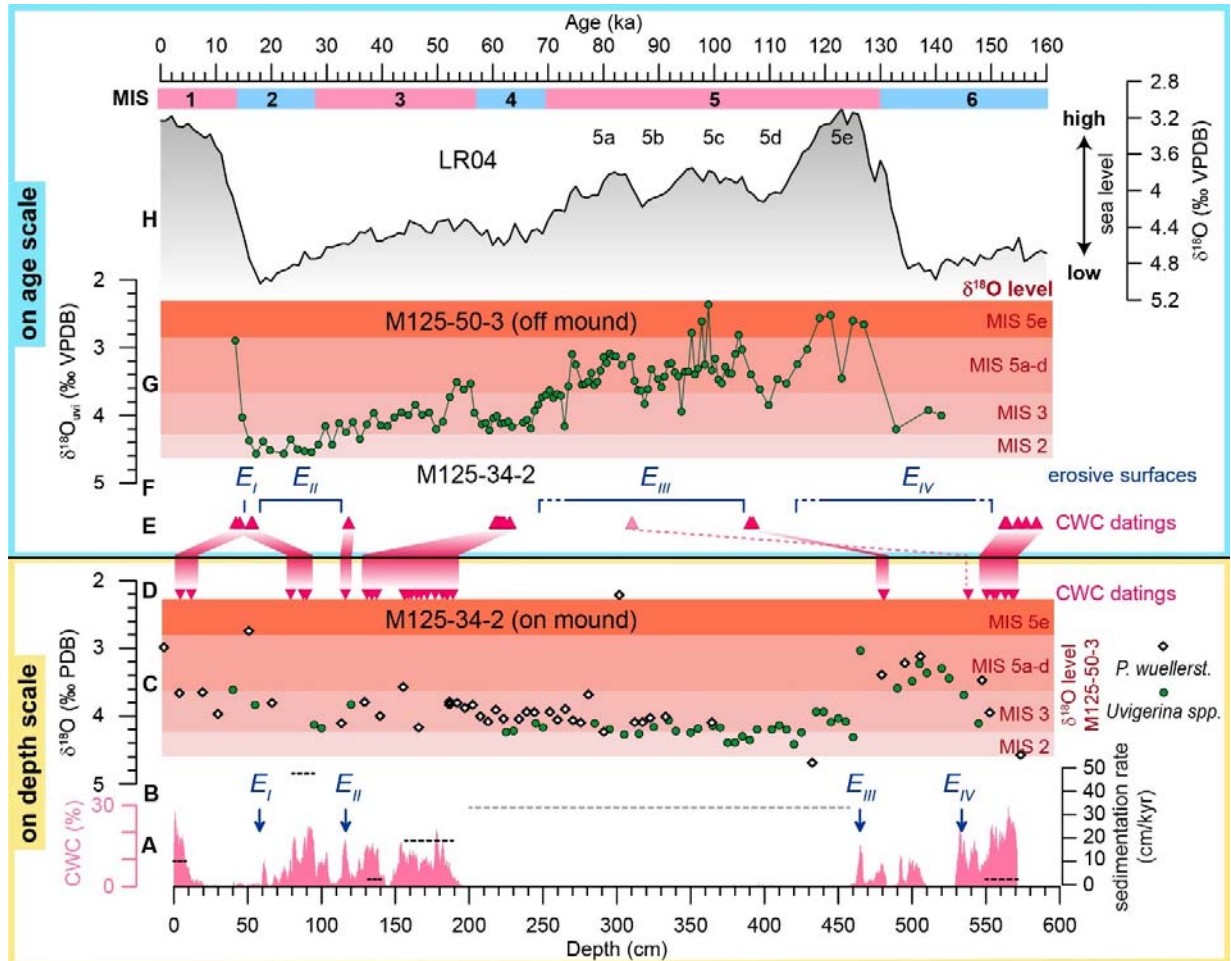


**Fig. 3** Refined age model of core M125-34-2 from Bowie Mound. Lower panel displays data from M125-34-2 on depth scale, comprising (A) CWC abundances (%; magenta) based on CT-scanning and accumulation rates for both, the Th/U-dated CWC bearing intervals (black dashed lines; cf. Raddatz et al., 2019) and the CWC-barren interval between 200–465 cm (grey dashed line); (B) blue arrows denote erosive horizons (hiatuses) $E_I$–$E_{IV}$ (cf. Fig. 2); (C)

benthic $\delta^{18}O$ data from *Uvigerina* spp. (green dots) and *P. wuellerstorfi* (white diamonds; elevated by +0.47 to adjust for the interspecies offset to *Uvigerina* spp. after Marchitto et al., 2014); red horizontal shadings indicate reference $\delta^{18}O$ levels for the respective Marine Isotope Stages (MIS) taken from off-mound core M125-50-3 (see panel G); (D) red triangles mark Th/U datings. The upper panel displays data in the time domain, with (E) Th/U dates of M125-34-2; (F) the inferred duration of hiatuses reflected by erosive horizons $E_I$–$E_{IV}$ of core M125-34-2; (G) the $\delta^{18}O$ record on

*Uvigerina* spp. on off-mound core M125-50-3 with red horizontal shadings illustrating the $\delta^{18}O$ level during different isotope stages as used in panel (C); (H) the LR04 benthic stack (Lisiecki and Raymo, 2005) as an approximation for eustatic sea-level changes; Marine Isotope Stages (MIS) are indicated by pink and blue bars signifying relatively warm and cold periods, respectively.


## 5. Results and Discussion

In the following we discuss if and how changes in environmental parameters might have enhanced or prohibited CWC growth at Bowie Mound, focusing on the role of intermediate water-mass variability and varying terrigenous sediment supply. We argue that the most dominant environmental factor for triggering CWC growth was elevated river run-off during periods of strong monsoonal rainfall in the coastal hinterland, which provided nutrients and organic matter that enhanced the food supply of CWC colonies.


### 5.1 Drivers and inhibitors of CWC growth at Bowie Mound

### 5.1.1 Intermediate water mass properties and hydraulic dynamics

Intermediate water-mass properties may have played a crucial role for the development of Bowie Mound. First, Bowie Mound lies within the AAIW, which probably boosted CWC growth at the East Brazilian slope. Secondly, nutrients and POC typically concentrate within nepheloid layers at water mass boundaries and provide a prolific food source for CWCs (Mienis et al., 2007; Dullo et al., 2008; Raddatz et al., 2014; Rüggeberg et al., 2016; Magill et al., 2018). In the case of Bowie Mound, it is possible that enhanced production (Pahnke and Zahn, 2005; Pahnke et al., 2008) and/or

nutrient-enrichment (e.g. increasing phosphate and nitrate concentration) of the AAIW (Poggemann et al., 2017) during phases of weak Atlantic Meridional Overturning Circulation (AMOC) could have triggered its episodic formation phases. A more prominent AAIW likely also strengthened internal waves hitting the slope at the AAIW/SACW boundary in the Campos Basin (Viana et al., 1998; Viana, 2001) and thus fueled the nepheloid layer by enhanced resuspension.

Here, we use the benthic $\delta^{13}$C obtained on M125-34-2 as an indicator of the relative contribution of $^{13}$C depleted southern-sourced intermediate water masses (Kroopnick, 1985; Curry and Oppo, 2005). As we had to use different species (shallow infaunal *Uvigerina* spp. and epibenthic *P. wuellerstorfi*), we combined both records. To adjust for intra-species offsets, we followed the approach of Kaboth et al. (2017) and normalized the $\delta^{13}$C values of each species to the respective mean and standard deviation. The resulting normalized data exhibits a considerable scatter in the

*Uvigerina* spp. record due to the pooling of different species, and hence only allows for a discussion of major shifts in the isotopic composition. However, the $\delta^{13}$C data of *Uvigerina* spp. and *P. wuellerstorfi* do not provide compelling evidence for distinctly depleted values during phases of CWC growth compared to CWC-barren intervals (Fig. 4). Although $\delta^{13}$C$_{Uvi}$ might be influenced by isotopic variations of dissolved inorganic carbon of pore-water (Zahn et al., 1986) and inter-species offsets (Rathburn et al. 1996; Theodor et al., 2016), we nevertheless consider it as appropriate

for reconstructing major changes in the bottom-water signature. Even when only considering $\delta^{13}$C$_{Plan}$, there are no apparent systematic difference between CWC-bearing and CWC-barren intervals. The continuous, monospecific $\delta^{13}$C$_{Uvi}$ record obtained on off-mound core M125-50-3 likewise lacks negative excursions during times of CWC growth at Bowie Mound (Fig. 5C). Hence, it appears that changes in the nutrient or organic matter content of the AAIW as observed during the last deglaciation (Poggemann et al., 2017) did not significantly affect CWC growth.


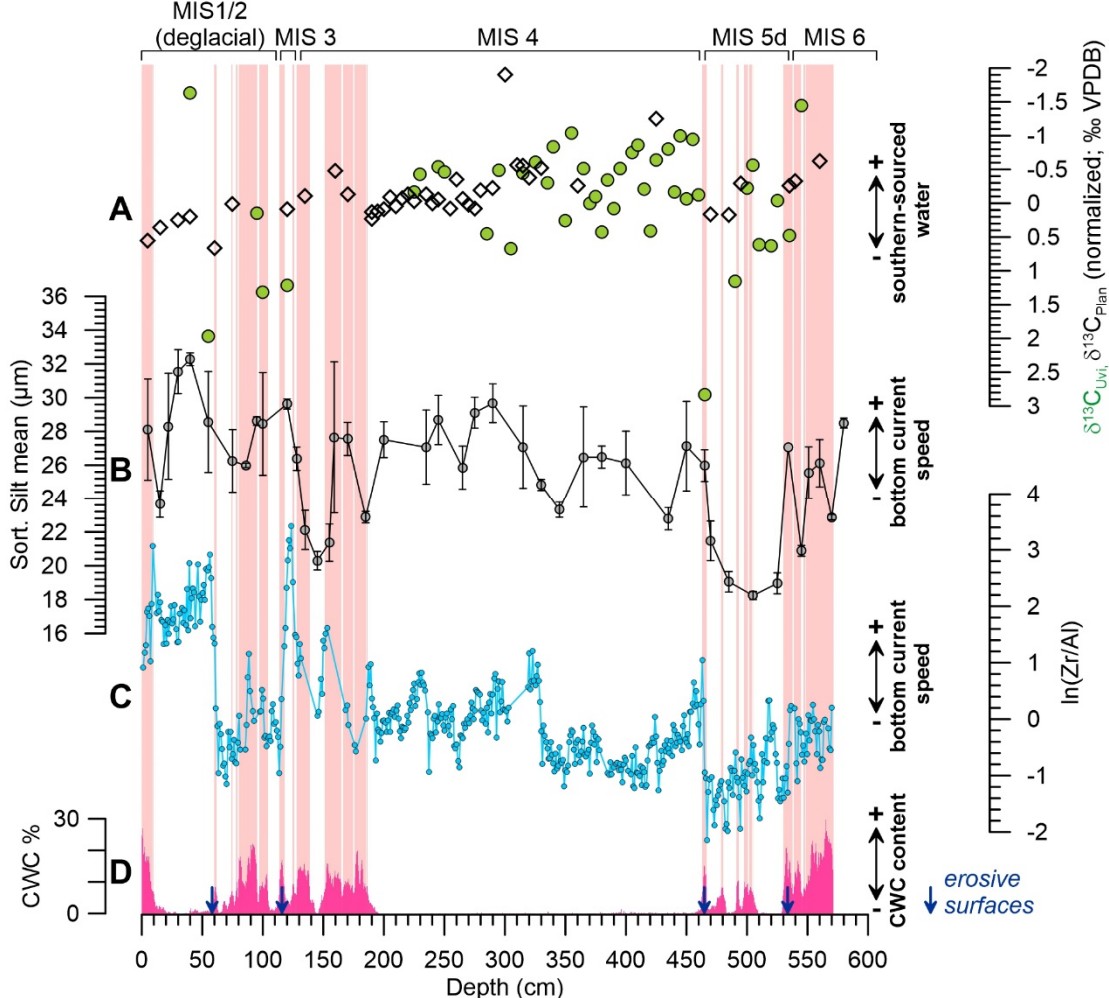

**Fig. 4** Proxies of intermediate water mass variability *vs.* phases of mound formation obtained on core M125-34-2. (A) normalized δ¹³C of *Uvigerina* spp. (green dots) and *P. wuellerstorfi* (white diamonds) as a proxy for bottom water-mass origin, (B) mean sortable silt ($\overline{SS}$) reflecting bottom current speed, (C) XRF-derived sedimentary ln(Zr/Al) ratio which are dependent on bottom current speed (Bahr et al., 2014) and advection of fine-grained material from the nepheloid layer, and (D) CWC abundances based on CT-scanning. Phases of CWC proliferation appear to require background state of high hydrodynamics conditions (elevated ln(Zr/Al) and $\overline{SS}$) but do not show an influence of deep-water mass variability (δ¹³C). Light red shadings denote intervals with coral contents >7 % used for discriminant analysis; blue arrows indicate erosive unconformities.

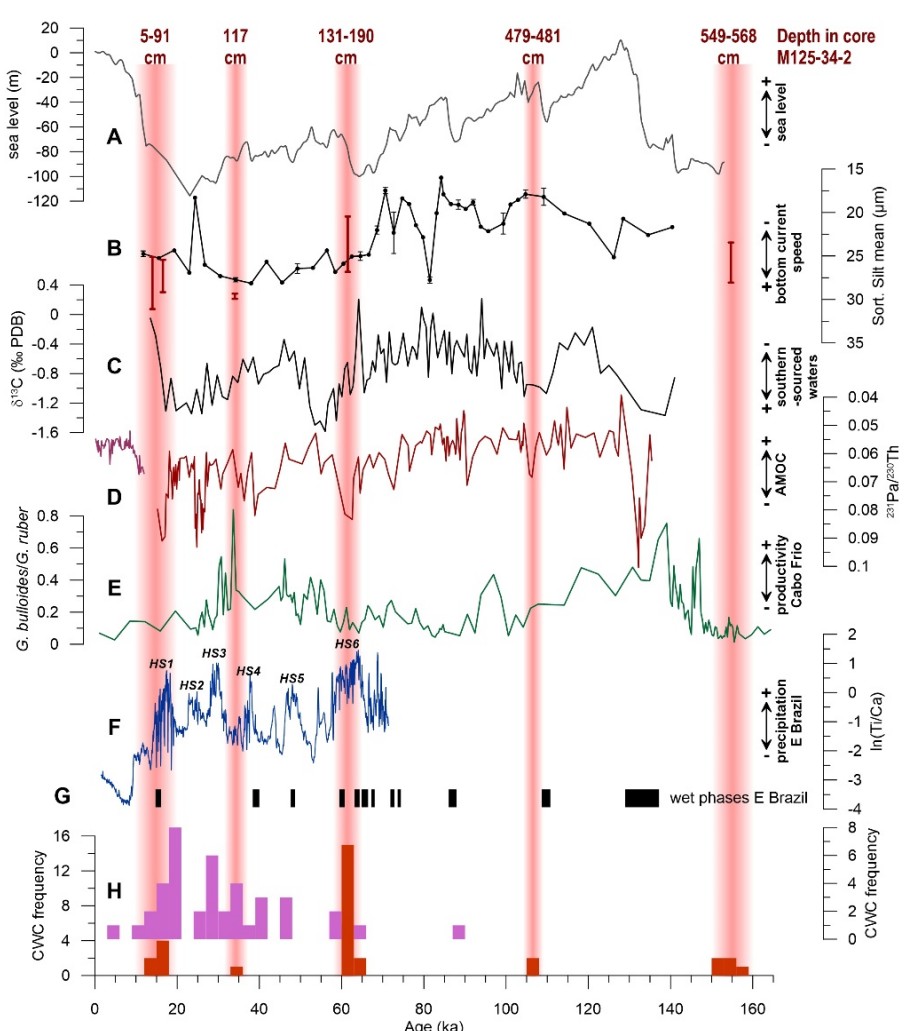

**Fig. 5** Frequency of CWC occurrences in the time domain at Bowie Mound core M125-34-2 and adjacent cores from the south-eastern Brazilian Margin in the paleoclimatic context. (A) global sea level (Grant et al., 2012); (B) mean sortable silt in off-mound core M125-50-3, range of sortable silt values of CWC-bearing intervals in core M125-34-2 are indicated by red bars; (C) benthic $\delta^{13}$C of *Uvigerina* spp. in off-mound core M125-50-3 reflecting the portion of southern-sourced waters (AAIW) at the south-eastern Brazilian Margin; (D) variability of Atlantic Meridional Overturning Circulation (AMOC; dark red line: Böhm et al., 2015; purple: Lippold et al., 2019); (E) upwelling intensity at Cabo Frio reflected by the ratio between upwelling-related planktonic foraminifer *Globigerinoides bulloides* and oligotrophic *Globigerinoides ruber*; (F) ln(Ti/Ca) ratio reflecting terrigenous input to the eastern Brazilian margin off the Sao Francisco River (core M125-95-3; Campos et al., 2019); (G) growth periods of travertine and speleothems in E Brazil during anomalous wet periods, indicated by black bars (Wang et al., 2004); (H) frequency of CWC Th/U dates per 3 kyrs bins at Bowie Mound (core M125-34-2; red bars) and in cores at the south-eastern Brazilian Margin (violet) (Mangini et al., 2010; Ruckelshausen, 2013) (Fig. 1). Note the good match between CWC occurrences and enhanced monsoonal activity on the continent (correlation between ln(Ti/Ca) and CWC frequency: r = 0.56, p = 0.02; computed using the SurrogateCor function of the R-package "astrochron"; Meyers, 2014). Red bars indicate periods of enhanced CWC growth at Bowie Mound, with the respective depths in core M125-34-2 annotated.

While nutrient and/or food delivery by the AAIW seems to have had an insignificant effect on CWC growth at Bowie Mound, changes in the hydrodynamics regime affecting the depth and strength of the nepheloid layer might have had an impact of CWC proliferation. Hydrodynamic conditions at core M125-34-2 can be reconstructed by using the variation in $\overline{SS}$, a well-established proxy for bottom current speed reflected by the mean grain size of the 10-63 μm fraction (McCave et al., 1995), and the sedimentary ln(Zr/Al) ratio (Fig. 4). The latter proxy follows the rationale that heavy minerals accumulate relative to aluminosilicates when the bottom current flow speed is high (e.g., Turnewitsch et al., 2004; Bahr et al., 2014; Miramontes et al., 2019). The significant correlation (r=0.49; p<0.05) of both proxies therefore predominantly reflects the hydraulic regime at Bowie Mound, despite certain intervals where both parameters deviate (e.g. between 60 and 115 cm with high $\overline{SS}$ and low ln(Zr/Al) values), which might be due to small-scale hydraulic effects created by the CWC branches themselves (Mienis et al., 2019). Based on both hydrodynamic proxies, CWC at Bowie Mound tend to accumulate during intervals with elevated flow speed. Phases of low flow speed are accompanied by low CWC abundances (e.g., at 140–150 cm and 465–550 cm), in line with the notion that active bottom currents play a significant role in distributing nutrients and food towards CWC colonies (e.g., Thiem et al., 2006; Dorschel et al., 2007; Davies et al., 2009; Raddatz et al., 2011). High current speeds will also increase sediment supply, thereby increasing accumulation rates due to the baffling capacity of CWC. Our data, however, suggest that a relatively high flow speed does not necessarily lead to CWC growth as demonstrated by the extended CWC-free section between 200 and 460 cm, where ln(Zr/Al) and $\overline{SS}$ reach high levels and at 15–60 cm where no CWC are preserved despite high TOC accumulation. Hence, the absence of CWC despite persistently high bottom current speeds during most of the glacial intervals MIS 3 and 4 indicate that environmental drivers other than intermediate water-mass variability must have played an important role for triggering CWC proliferation on the Brazilian Margin.

### 5.1.2 Terrestrial and marine organic matter supply

Having a sufficient food supply to Bowie Mound is a prerequisite for enhanced CWC growth. To assess the potential impact of enhanced POC and/or DOC supply as an incentive for enhanced CWC growth we compared TOC measurements to CWC abundances (Fig. 6). It appears that CWC abundance is highest during intervals of elevated organic matter content, while the long CWC-barren interval between 200 and 460 cm is characterized by relatively low TOC contents, thus stressing the importance of TOC as a prerequisite for mound aggregation. While statistically significant, the down-core pattern of CWC abundances and TOC (r = 0.55; p < 0.05) also indicates that this correlation is not straight forward; an example is the coral-free, high-TOC interval between 15 and 60 cm (Fig. 6). However, as suggested in the previous section, there are multiple factors necessary for stimulating coral growth at Bowie Mound.

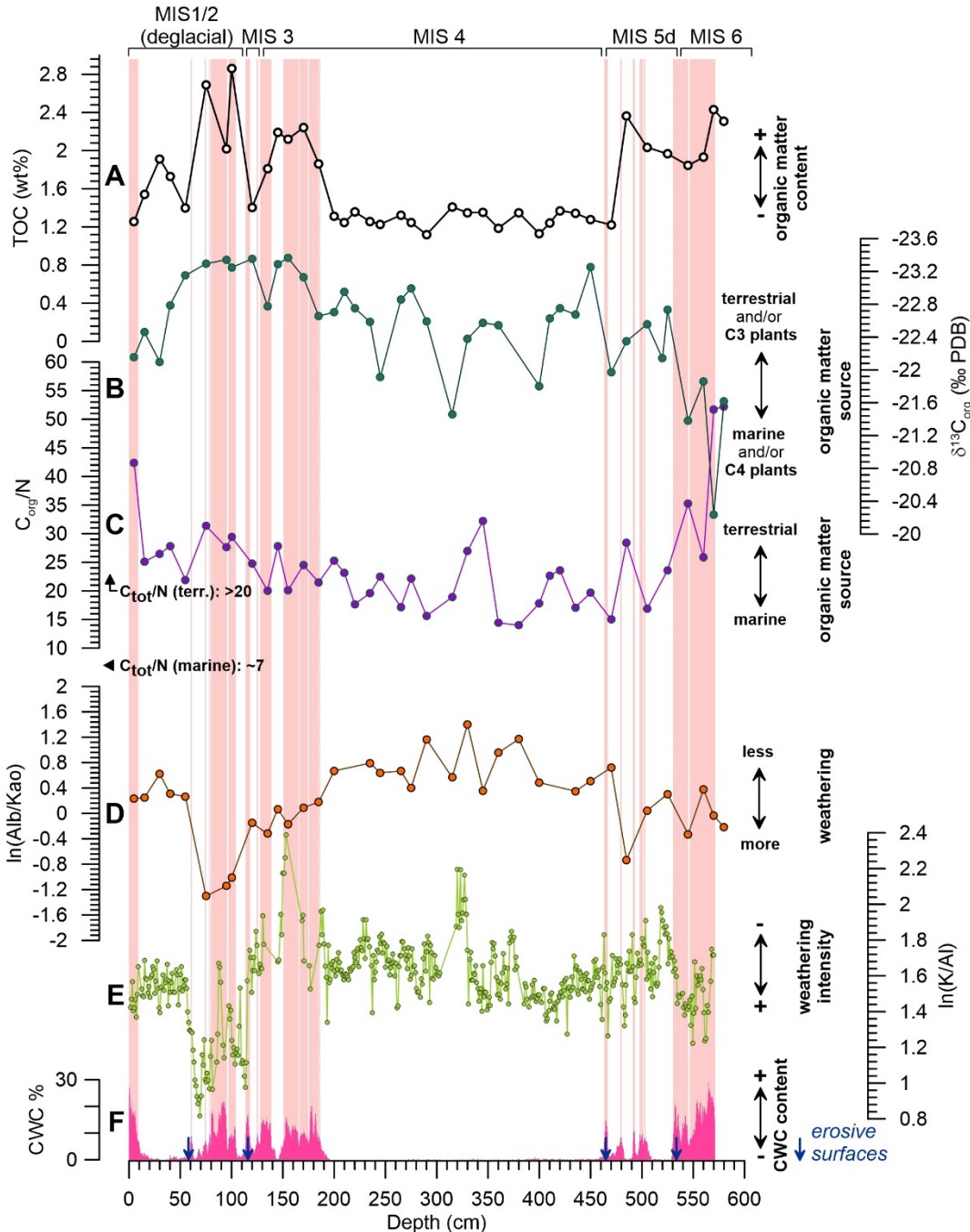

**Fig. 6** Organic-matter accumulation and origin at Bowie Mound core M125-34-2 in context with continental hydroclimate and CWC occurrences. (A) Total organic carbon (TOC, white dots) reflecting organic-matter accumulation; (B) and (C) $\delta^{13}C$ and $C_{org}/N_{tot}$ ratio of organic matter as measures for terrestrial *vs.* marine organic matter input, respectively. Marine and terrestrial endmembers are indicated (Holtvoeth et al., 2003); (D) and (E) XRD-derived ln(Albite/Kaolinite) and XRF-derived ln(K/Al) ratios, respectively, as indicators of the weathering intensity in the hinterland, reflecting the strength of the continental hydrological cycle; and (F) CWC abundances based on CT-scanning. Note that high CWC abundances fall into intervals of high TOC and increased weathering due to an intensified continental hydrological cycle. Light red shadings denote intervals with coral contents >7 % used for discriminant analysis; blue arrows indicate erosive unconformities.

To investigate the origin of the organic matter within the sediment matrix we analyzed its $C_{org}/N_{total}$ ratio and stable carbon isotopic composition ($\delta^{13}C_{org}$). Marine organic matter has lower $C_{org}/N_{total}$ ratios (~6.6) compared to terrestrial organic matter (>20) (Holtvoeth et al., 2003). Organic matter in core M125-34-2 exhibits $C_{org}/N_{total}$ ratios of 14 to 52 (Fig. 6), indicating that land-derived material was a dominant source of organic matter throughout the record. Intervals of high CWC abundances are characterized by elevated $C_{org}/N_{total}$ ratios, namely at the top and at the base of the core with ratios >40 indicating an overwhelming contribution of terrigenous matter. Notably, in line with the upwelling center off Cabo Frio being confined to the shelf without reaching northward to the slope where Bowie Mound is situated (Albuquerque et al., 2014, 2016), upwelling derived organic matter with typical marine $C_{org}/N_{total}$ ratios below 6.5 (Albuquerque et al., 2014) apparently played a subordinate role. A significant terrigenous admixture of terrestrial organic matter is also indicated by the relatively low $\delta^{13}C_{org}$ (-23.2 to -20.2 ‰) as terrigenous organic material has a more depleted signature (-27 ‰) compared to typical marine $\delta^{13}C_{org}$ values (-19 ‰) (Holtvoeth et al., 2003). In this regard, high $\delta^{13}C_{org}$ values during CWC-bearing intervals in the upper 50 cm and below 549.5 cm can be interpreted as periods of enhanced marine productivity, which contradicts contemporaneous $C_{org}/N_{total}$ ratios of as high as 52. This conflicting evidence might be resolved by interpreting the high $\delta^{13}C_{org}$ values as being influenced by enhanced input of POC from C4 plants (typically grasses), which are characterized by an endmember of ca. -12 ‰ (Holtvoeth et al., 2003). Palynological evidence indeed point to the establishment of grassland biomes in the catchment of the Paraiba do Sul during the last glacial as a result of generally drier conditions (Behling et al., 2002). This is in accordance with the presumed deposition of the intervals 0–50 cm and 549.5–568 cm during the last deglaciation and MIS 6, respectively (Fig. 5).

### 5.1.3 Influence of the continental hydrological cycle

A major source of terrigenous material and thus a potent organic-matter source for the Brazilian Margin are rivers draining the densely vegetated hinterland, especially the Paraiba do Sul (Fig. 1). To investigate if enhanced riverine input due to more humid conditions in the hinterland increased terrestrial organic-matter supply to Bowie Mound, we studied the mineralogical and geochemical composition of the terrigenous fraction of the sediment. Changes in the water availability in the hinterland should impact the degree of weathering and thus leave an imprint in the mineralogical composition of the terrigenous sediments. The composition of the non-carbonaceous mineral phases is typical for soils that underwent different degrees of chemical weathering, comprising intermediate weathering products such as hydrobiotite (11.5 %, up to 24 %) (Coleman et al., 1963; Wilson, 1970; Meunier and Velde, 1979), as well as typical constituents of soils that have been deeply weathered under tropical humid conditions such as kaolinite (7.9 % up to 15.3 %), gibbsite (on average 21.7 %) and Ca, K, Al-rich Zeolite (6.7 %) (e.g. Weaver, 1975; Hughes, 1980; Ibrahim and Hall, 1996; Furian et al., 2002) (cf. Appendix Table A). When comparing minerals typical for residual soils such as kaolinite, gibbsite, and zeolite with feldspar and mica, both groups are anti-correlated (Table XRD) and exhibit distinct fluctuations throughout the core (cf. the albite *vs*. kaolinite ratio is depicted in Fig. 6). Relatively high abundances of weathering residuals like kaolinite are particularly present in the CWC-bearing interval between ca. 60 and 200 cm and to a lesser extent below 560 cm. This XRD-based data is also partly supported by the high-resolution ln(K/Al) record obtained via XRF scanning, which is particularly low during the interval 60–120 cm and below 530

cm (Fig. 6). Low ln(K/Al) ratios are consistent with high kaolinite contents reflecting periods of strong K-removal from soils due to chemical weathering during humid conditions in the hinterland. As the $C_{org}/N_{total}$ ratio is also elevated during periods of low ln(K/Al) ratios and high kaolinite contents (Fig. 6), it can be inferred that high precipitation in the hinterland enhanced chemical weathering and increased terrigenous organic matter transport to the continental slope. Stronger chemical weathering would have also enhanced the input of Fe to the continental slope (Govin et al., 2012) leading to fertilization of the surface waters, which likely further invigorated surface productivity and increased the food supply for CWC at Bowie Mound.

**Table XRD:** Correlation between terrigenous mineral phases in core M125-34-2. Positive correlations are marked in red (for r>0.4), negative correlations (for r<0.4) in blue.

| | Quartz | Gibbsite | Kaolinite | Muscovite | Hydrobiotite | Microcline | Zeolite |
|---|---|---|---|---|---|---|---|
| **Quartz** | | | | | | | |
| **Gibbsite** | -0.07 | | | | | | |
| **Kaolinite** | -0.52 | -0.09 | | | | | |
| **Muscovite** | 0.35 | 0.45 | -0.31 | | | | |
| **Hydrobiotite** | -0.34 | -0.62 | 0.55 | -0.69 | | | |
| **Microcline** | 0.49 | -0.26 | -0.43 | -0.15 | -0.02 | | |
| **Zeolite** | -0.16 | 0.68 | 0.00 | 0.62 | -0.43 | -0.51 | |
| **Albite** | 0.65 | -0.07 | -0.76 | 0.47 | -0.44 | 0.35 | -0.07 |

To objectively evaluate the relative importance of terrestrial organic matter supply over changes in the hydraulic conditions in influencing CWC proliferation at Bowie Mound we performed a discriminant analysis over n=34 samples using (i) ln(Alb/Kao) as a weathering proxy, (ii) $\overline{SS}$ for bottom current speed, (iii) $\delta^{13}C_{org}$, and (iv) $C_{org}/N_{total}$ for organic matter provenance as predictors for the occurrences of CWC abundances above 7 % (reflecting representative CWC accumulations). Note that the $\delta^{13}C$ of benthic foraminifera had not been included due to the high scatter of the data. The restriction on those four parameters was also done to avoid unreliable results by over-prediction. The outcome of the discriminant analysis yields a correct classification of 76 % with highest skills in ln(Alb/Kao) (loading of -0.68) and $C_{org}/N_{total}$ (+0.58), while loadings for $\overline{SS}$ (+0.06) and $\delta^{13}C_{org}$ (0.00) are insignificant (see Appendix Table B for details). This is consistent with peak CWC abundances during phases of strong hydrological activity, as evidenced by intensified chemical weathering (i.e. low ln(Alb/Kao) ratios), which enhanced the input of terrigenous organic matter (high $C_{org}/N_{total}$) to Bowie Mound and directly or indirectly (via surface water fertilization) stimulated CWC proliferation.

**5.2 CWC at Bowie Mound in the paleoclimatological context**

As discussed above, phases of enhanced CWC proliferation at Bowie Mound occurred in parallel to periods of increased terrigenous POC and/or DOC input due to enhanced run-off. The most distinct CWC proliferation phases in fact took place during phases of anomalously strong monsoonal precipitation during the pronounced Heinrich Stadials (HS) 1 and 6 as well as (within age model uncertainties) also during the shorter and less severe humid phases corresponding to HS 2–5 (Fig. 5F, G). The prominent growth phase during HS is in agreement with published CWC

occurrences along the south-eastern Brazilian Margin from the same time frame (Mangini et al., 2010; Ruckelshausen, 2013) (Figs. 1, 5G), indicating that our results are representative for a larger geographic area. The slow overturning circulation during HS indicated by high $^{231}$Pa/$^{230}$Th ratios (Fig. 5D; McManus et al., 2004; Böhm et al., 2015) resulted in enhanced precipitation over south-eastern and eastern Brazil (Waelbroeck et al., 2018) due to heat accumulation in the southern hemisphere and intensification of the South Atlantic Convergence Zone (Fig. 1; Stríkis et al., 2015). Humid phases during HS in otherwise dry eastern Brazil are documented by growth phases of speleothems and travertines (Fig. 5G), increased terrigenous input (core M125-95-3; Fig. 5F) (Campos et al., 2019), and a slight expansion of forest cover (Gu et al., 2018). It is possible that the riverine suspension load from other rivers in eastern Brazil was advected southward by the BC and added to the enhanced terrigenous load from rivers adjacent to Bowie Mound (mostly the Paraiba do Sul). Due to the baffling capacity of framework-building CWCs (Mienis et al., 2007; Huvenne et al., 2009; Titschack et al., 2015), the additional sedimentary input would have aided mound formation. This nutrient and organic-rich suspension potentially also enhanced marine surface productivity and thus directly and/or indirectly boosted food supply to the CWC. As freshwater admixture to the SMW and SACW may increased water column stratification and hence the density contrast to the AAIW, the concentration of sediment and food particles at the nepheloid layer may also have been more pronounced. The link between Monsoonal activity and CWC growth proposed here is in line with studies from the western Mediterranean Sea (Fentimen et al., 2020), the Gulf of Cadiz (Wienberg et al., 2010) and the tropical eastern Atlantic off Angola (Hanz et al., 2019) which all inferred that terrestrial input via dust or fluvial run-off can ultimately fuel CWC colonies.

Notably, HS occurred during phases of intermediate and low sea level, which facilitated the bypass of sediment across the shelf and thus allowed for an efficient supply of terrestrial organic matter to Bowie Mound (Fig. 5A). As discussed in Raddatz et al. (2020), during glacials, when sea level was up to 120 m lower than present (Waelbroeck et al., 2002; Rohling et al., 2014), water-mass boundaries may have been forced to migrate downslope. Such a displacement of water-mass boundaries would have moved the SACW/AAIW interface and the corresponding nepheloid layer from its present position at ~500 m closer to the depth of Bowie Mound (~860 m), aiding the concentration and dispersal of food along the slope towards the CWC colonies. This suspected influence of sea level on the bottom-current dynamics at the depth of Bowie Mound is in fact evident from a sortable silt record obtained from the off-mound site M125-50-3, which shows high (low) current speeds during low (high) sea level (Fig. 5B).

Hence, we argue that a dynamic hydraulic regime along with a pronounced nepheloid layer was required for CWC to flourish at Bowie Mound (Mienis et al., 2007). These prerequisite conditions were present throughout glacial periods, during which the development of wide-spread glacial unconformities and extensive drift bodies along the south-east Brazilian margin (Viana et al., 1998; Viana, 2001) are evidence for such intense bottom-current activity. This is notably different from the slope south of the Abrolhos Bank where reference core M125-50-3 is situated. Here, contouritic sediments and distinct hiatuses are largely absent in water depths affected by the AAIW and SACW, which testifies to less dynamic hydrological conditions (Bahr et al., 2016), potentially responsible for the lack of CWC mounds at this location. At the same time, $\overline{SS}$ and ln(Zr/Al) data from both Bowie Mound core M125-34-2 (Fig. 4B) and off-mound core M125-50-3 (Fig. 5B) indicate that bottom-current speed was not anomalously high during HS relative to the glacial background level. Hence, the distinct, pulse-like CWC growth phases at Bowie Mound could have been initiated by enhanced nutrient and organic matter supply from land as evidenced by the high $C_{org}$/N ratio of the organic matter.

Surface water fertilization by the increase in terrigenous organic matter may have also improved marine primary productivity and contributed to higher export production, further fueling CWC growth. As discussed before, marine productivity caused by from upwelling on the shelf of Cabo Frio was apparently of minor importance. Based on planktic foraminiferal assemblages (Lessa et al., 2019), enhanced upwelling occurred between HS 3 and 4, at around 35 kys, when a small peak of CWC abundances occurs (Fig. 5G). However, during all other instances of CWC occurrences, upwelling at Cabo Frio was relatively low and likely did not reached as far north as Bowie Mound (Albuquerque et al., 2014, 2016).

We hence infer that CWC proliferation at Bowie Mound occurred simultaneously with an enhanced delivery of terrestrial organic matter towards the continental slope under glacial boundary conditions of low sea level and enhanced hydrodynamic activity along the slope. This implies the direct utilization of terrestrial organic matter by the corals and thus clearly stresses the necessity for future in-depth studies of the food preferences of *S. variabilis*.

**Conclusions**

Here we present a comprehensive multi-proxy study of a CWC-bearing core retrieved off south-eastern Brazil with the aim to assess the relative importance of different environmental factors that supported and/or prohibited CWC proliferation and coral mound formation. We find that intervals of high CWC abundance are primarily related to millennial-scale high latitude cold events (Heinrich Stadials) that were characterized by major reconfigurations of deep-ocean circulation and enhanced monsoonal precipitation in eastern Brazil. The dominance of terrigenous- over marine-derived organic matter during phases of fast CWC proliferation indicate that strong run-off enhanced the input of nutrients and food to coral mounds on the continental slope. Intensified hydrodynamic conditions at the water depth of Bowie Mound during sea level lowstands thereby provided the necessary background conditions for an efficient dispersal of nutrient and food supply towards the upper and mid slope. Thus, this study presents a prime example of the intimate coupling between continental hydroclimate and ecological changes in the deep ocean.

**Appendices**

**Appendix Table A:** Major non-carbonaceous mineral phases in core M125-34-2 derived from Rietveld analyses of X-ray diffractometry.

| Depth (cm) | Quartz | Gibbsite | Kaolinite | Muscovite | Hydrobiotite | Microcline | Zeolite | Albite |
|---|---|---|---|---|---|---|---|---|
| | (%) | (%) | (%) | (%) | (%) | (%) | (%) | (%) |
| 5 | 8.5 | 20.2 | 5.7 | 5.6 | 7.8 | 7.0 | 6.7 | 7.2 |
| 15 | 8.0 | 19.1 | 7.8 | 5.5 | 16.1 | 4.6 | 6.7 | 10.0 |
| 30 | 6.9 | 18.7 | 6.5 | 2.1 | 10.1 | 6.3 | 4.2 | 12.1 |
| 40 | 7.6 | 20.3 | 7.7 | 3.4 | 17.1 | 9.3 | 3.8 | 10.5 |
| 55 | 10.4 | 18.4 | 8.7 | 4.0 | 14.1 | 3.7 | 5.0 | 11.3 |
| 75 | 4.6 | 20.0 | 15.3 | 4.8 | 20.2 | 2.1 | 6.5 | 4.2 |
| 95 | 4.3 | 17.6 | 12.0 | 4.4 | 21.7 | 4.8 | 5.9 | 3.8 |
| 100 | 4.8 | 16.4 | 12.6 | 5.6 | 24.0 | 3.2 | 6.2 | 4.6 |
| 120 | 9.7 | 15.5 | 10.3 | 4.4 | 18.2 | 6.2 | 4.7 | 8.9 |
| 135 | 8.3 | 24.5 | 9.9 | 9.4 | 12.9 | 4.3 | 6.7 | 7.2 |
| 145 | 7.1 | 24.1 | 9.0 | 5.1 | 17.4 | 3.3 | 7.1 | 9.6 |
| 155 | 7.8 | 25.9 | 10.2 | 11.5 | 7.8 | 4.6 | 7.9 | 8.6 |
| 170 | 7.6 | 22.3 | 9.0 | 6.1 | 17.6 | 4.0 | 7.1 | 9.8 |
| 185 | 9.2 | 25.1 | 7.4 | 11.5 | 6.0 | 6.2 | 7.4 | 8.9 |
| 200 | 10.2 | 21.3 | 6.2 | 6.2 | 12.5 | 7.9 | 6.6 | 12.2 |
| 235 | 9.2 | 20.6 | 5.4 | 13.3 | 10.7 | 3.6 | 7.4 | 12.0 |
| 245 | 9.5 | 22.6 | 6.5 | 13.2 | 5.7 | 5.8 | 6.8 | 12.4 |
| 265 | 8.1 | 22.8 | 6.5 | 9.9 | 9.8 | 6.4 | 7.4 | 12.7 |
| 275 | 9.2 | 23.5 | 7.5 | 12.9 | 6.9 | 5.4 | 7.0 | 11.3 |
| 290 | 9.6 | 21.6 | 4.0 | 10.7 | 9.7 | 6.9 | 6.3 | 13.0 |
| 315 | 10.4 | 21.9 | 6.3 | 12.8 | 7.1 | 5.0 | 6.7 | 11.1 |
| 330 | 9.3 | 19.8 | 4.0 | 11.4 | 8.7 | 5.1 | 6.4 | 16.3 |
| 345 | 7.7 | 19.7 | 8.3 | 11.9 | 10.2 | 4.9 | 7.1 | 11.9 |
| 360 | 6.9 | 19.4 | 5.5 | 9.0 | 10.0 | 5.8 | 6.3 | 14.3 |
| 380 | 8.3 | 23.1 | 4.5 | 13.7 | 5.2 | 3.2 | 7.2 | 14.5 |
| 400 | 7.0 | 21.2 | 6.2 | 7.6 | 12.9 | 6.8 | 6.5 | 10.1 |
| 435 | 7.9 | 23.1 | 8.3 | 12.6 | 6.2 | 3.1 | 6.8 | 11.8 |
| 450 | 7.8 | 23.1 | 7.0 | 12.0 | 7.4 | 3.8 | 7.4 | 11.7 |
| 470 | 8.3 | 18.6 | 6.4 | 11.8 | 14.6 | 4.6 | 7.5 | 13.2 |
| 485 | 6.3 | 27.4 | 12.2 | 12.3 | 5.5 | 2.8 | 8.5 | 5.8 |
| 505 | 5.5 | 23.7 | 9.5 | 12.5 | 6.2 | 2.8 | 7.7 | 9.9 |
| 525 | 5.7 | 24.6 | 7.4 | 10.2 | 11.2 | 3.6 | 8.0 | 10.1 |
| 545 | 4.4 | 23.1 | 8.4 | 4.4 | 10.3 | 3.0 | 7.4 | 6.0 |
| 560 | 6.5 | 26.3 | 5.3 | 6.8 | 8.1 | 3.8 | 7.1 | 7.8 |

| 570 | 5.8 | 22.0 | 7.3 | 6.0 | 10.5 | 3.5 | 6.4 | 7.0 |
|-----|-----|------|-----|-----|------|-----|-----|-----|
| 580 | 5.3 | 23.0 | 8.1 | 5.8 | 12.9 | 2.8 | 7.3 | 6.5 |

**APPENDIX Table B.** Result of discriminant analysis performed on detrended and normalized proxies obtained on core M125-34-2. Samples were divided into classes with CWC contents >7% (labelled "1") and <7% ("0"). Classes derived from discriminant analysis are displayed in the last column, with wrongly assigned classes marked in red.

| depth (cm) | ln(Alb/Kao) | $\overline{SS}$ (µm) | $\delta^{13}C_{org}$ (‰ PDB) | $C_{org}/N$ | CWC > 7% | inferred class |
|------------|-------------|----------------------|------------------------------|-------------|----------|----------------|
| 5 | 0.01 | 0.71 | 0.58 | 2.03 | 1 | 1 |
| 15 | 0.03 | -0.61 | 0.11 | 0.04 | 0 | 0 |
| 30 | 0.64 | 1.73 | 0.67 | 0.20 | 0 | 0 |
| 40 | 0.13 | 1.94 | -0.39 | 0.35 | 0 | 0 |
| 55 | 0.06 | 0.84 | -0.95 | -0.33 | 0 | 0 |
| 75 | -2.49 | 0.15 | -1.17 | 0.77 | 1 | 1 |
| 95 | -2.23 | 0.86 | -1.24 | 0.33 | 1 | 1 |
| 100 | -2.01 | 0.81 | -1.10 | 0.54 | 1 | 1 |
| 120 | -0.61 | 1.16 | -1.25 | 0.00 | 0 | 1 |
| 135 | -0.89 | -1.07 | -0.37 | -0.55 | 1 | 0 |
| 145 | -0.26 | -1.61 | -1.16 | 0.35 | 0 | 0 |
| 155 | -0.65 | -1.30 | -1.28 | -0.54 | 1 | 0 |
| 170 | -0.22 | 0.54 | -0.91 | -0.03 | 1 | 0 |
| 185 | -0.08 | -0.85 | -0.18 | -0.38 | 1 | 0 |
| 200 | 0.72 | 0.53 | -0.26 | 0.06 | 0 | 0 |
| 235 | 0.92 | 0.40 | -0.08 | -0.60 | 0 | 0 |
| 245 | 0.67 | 0.88 | 0.95 | -0.26 | 0 | 0 |
| 265 | 0.72 | 0.04 | -0.49 | -0.88 | 0 | 0 |
| 275 | 0.29 | 1.00 | -0.71 | -0.30 | 0 | 0 |
| 290 | 1.53 | 1.17 | -0.09 | -1.06 | 0 | 0 |
| 315 | 0.55 | 0.40 | 1.65 | -0.68 | 0 | 0 |
| 330 | 1.91 | -0.27 | 0.24 | 0.25 | 0 | 0 |
| 345 | 0.21 | -0.72 | -0.06 | 0.86 | 0 | 0 |
| 400 | 0.42 | 0.12 | 1.12 | -0.80 | 0 | 0 |
| 435 | 0.20 | -0.87 | -0.22 | -0.88 | 0 | 0 |
| 450 | 0.46 | 0.41 | -1.10 | -0.59 | 0 | 0 |
| 470 | 0.80 | -1.26 | 0.86 | -1.13 | 0 | 0 |
| 485 | -1.56 | -1.98 | 0.28 | 0.42 | 0 | 1 |
| 505 | -0.30 | -2.22 | -0.04 | -0.91 | 0 | 0 |
| 525 | 0.13 | -2.01 | -0.30 | -0.13 | 0 | 0 |
| 545 | -0.90 | -1.44 | 1.77 | 1.21 | 0 | 1 |
| 560 | 0.24 | 0.11 | 1.03 | 0.13 | 1 | 0 |
| 570 | -0.43 | -0.85 | 3.52 | 3.12 | 1 | 1 |
| 580 | -0.72 | 0.82 | 1.40 | 3.17 | 1 | 1 |

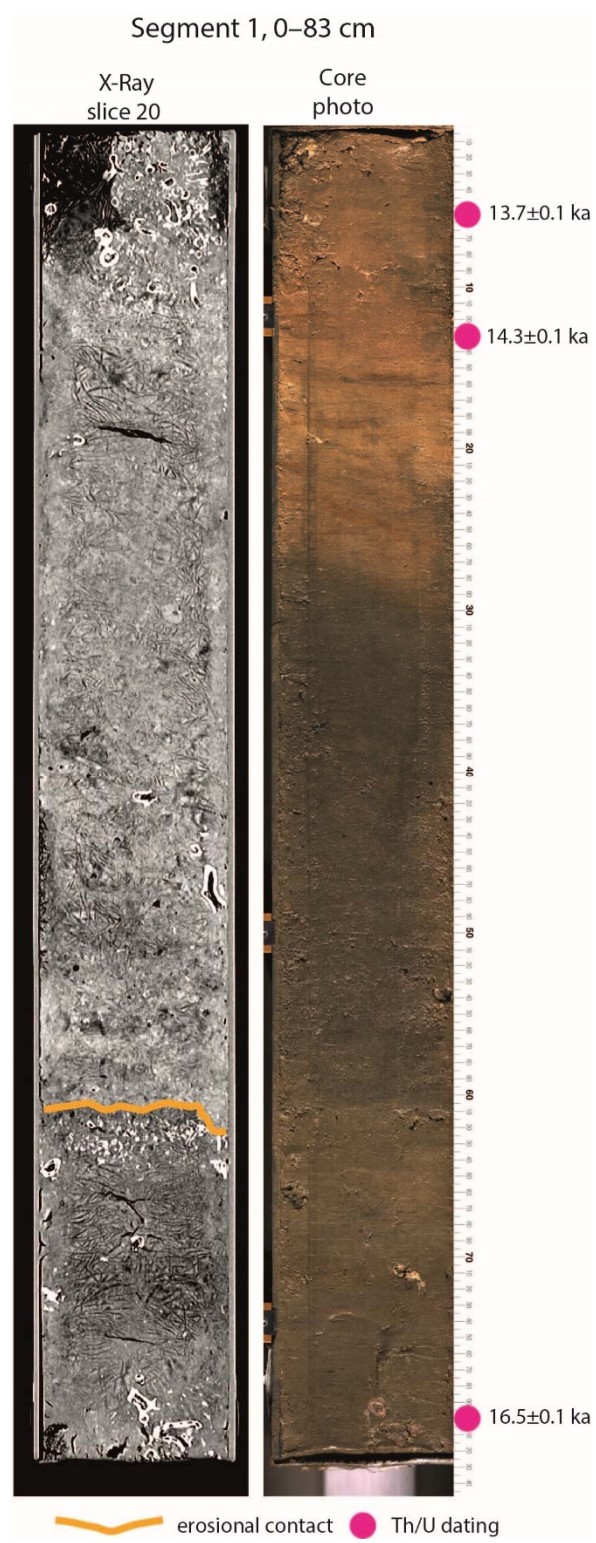

**Appendix Figure A:** CT image (left) and photography (right) of segment 0-83 cm of core M125-34-2. Erosional contacts are marked by thick black lines, position of Th/U datings with calibrated ages are indicated by magenta dots.

## Segment 2, 83–183 cm

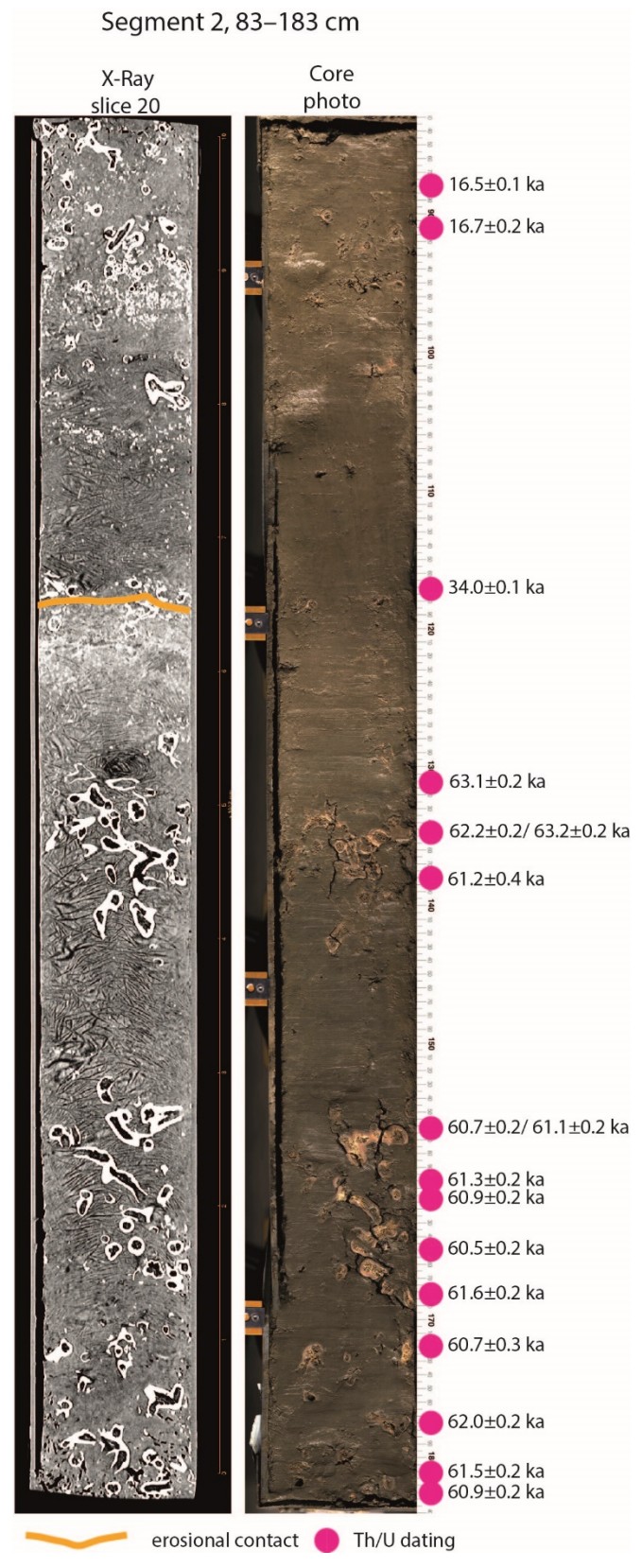

**Appendix Figure B**: CT image (left) and photography (right) of segment 83-183 cm of core M125-34-2. Erosional contacts are marked by thick black lines, position of Th/U datings with calibrated ages are indicated by magenta dots.

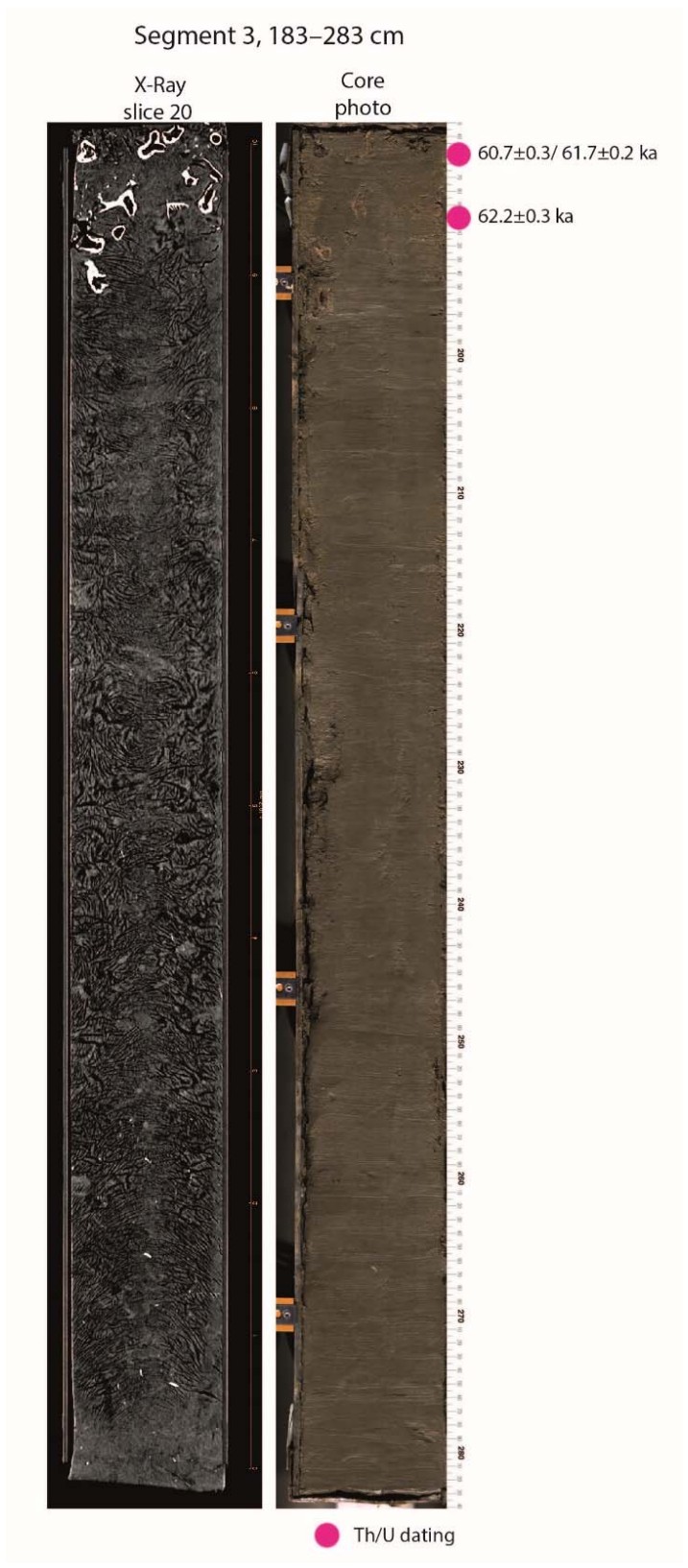

**Appendix Figure C:** CT image (left) and photography (right) of segment 183-283 cm of core M125-34-2. Position of
Th/U dating with calibrated age is indicated by a magenta dot.

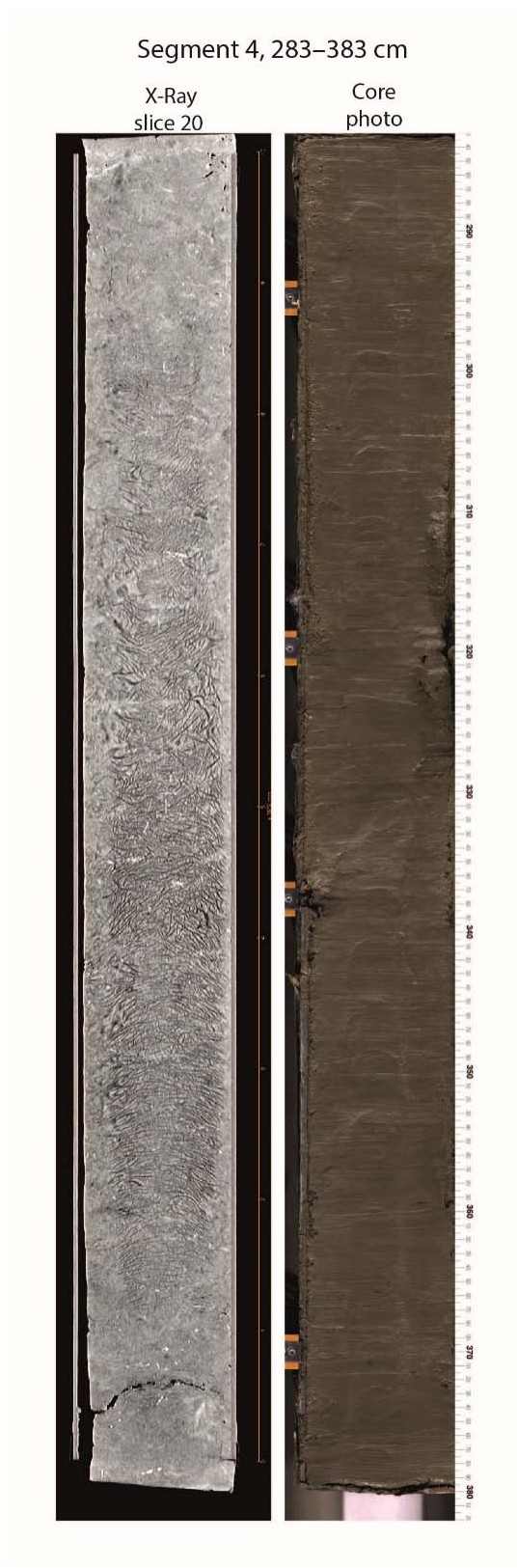

**Segment 4, 283–383 cm**

**Appendix Figure D**: CT image (left) and photography (right) of segment 283-383 cm of core M125-34-2.

Segment 5, 383–483 cm

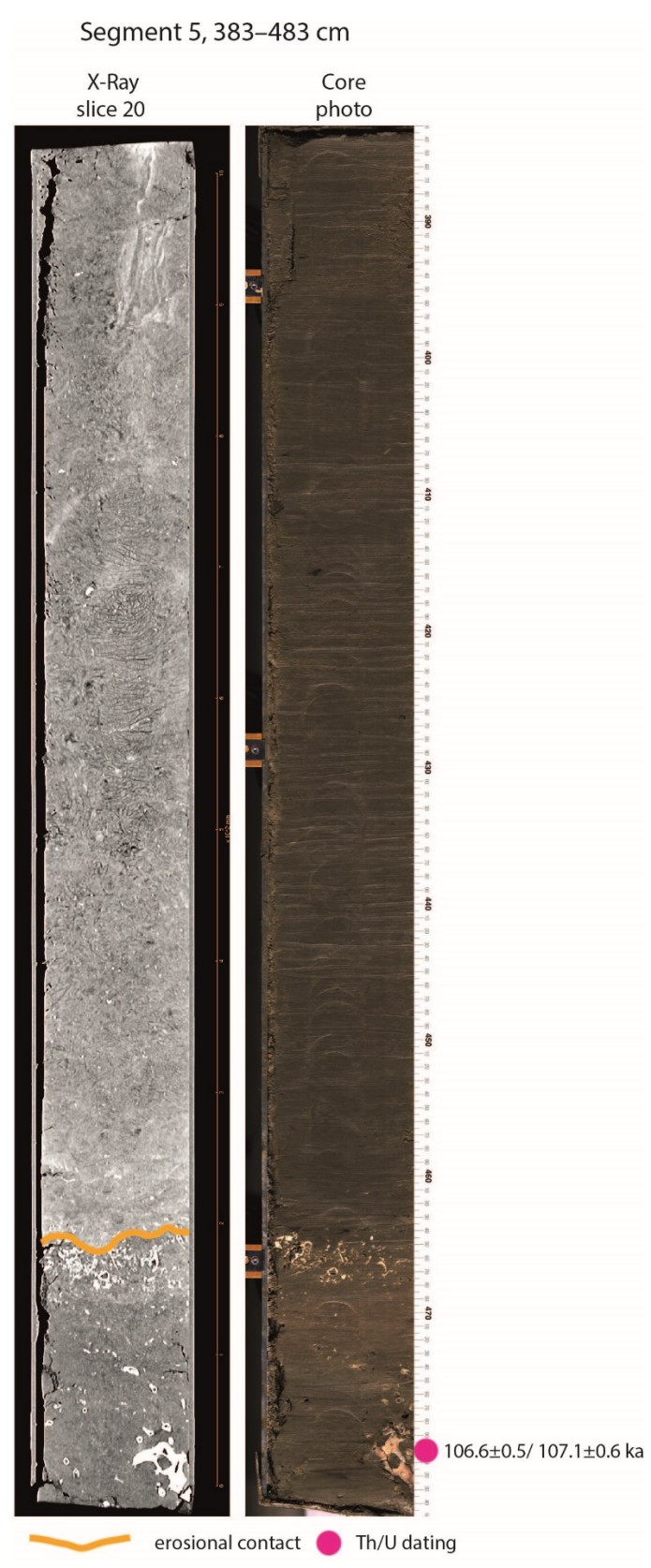

X-Ray
slice 20

Core
photo

106.6±0.5/ 107.1±0.6 ka

⌇ erosional contact  ● Th/U dating

**Appendix Figure E:** CT image (left) and photography (right) of segment 383-483 cm of core M125-34-2. Erosional contacts are marked by thick black lines, position of Th/U dating with calibrated age is indicated by a magenta dot.

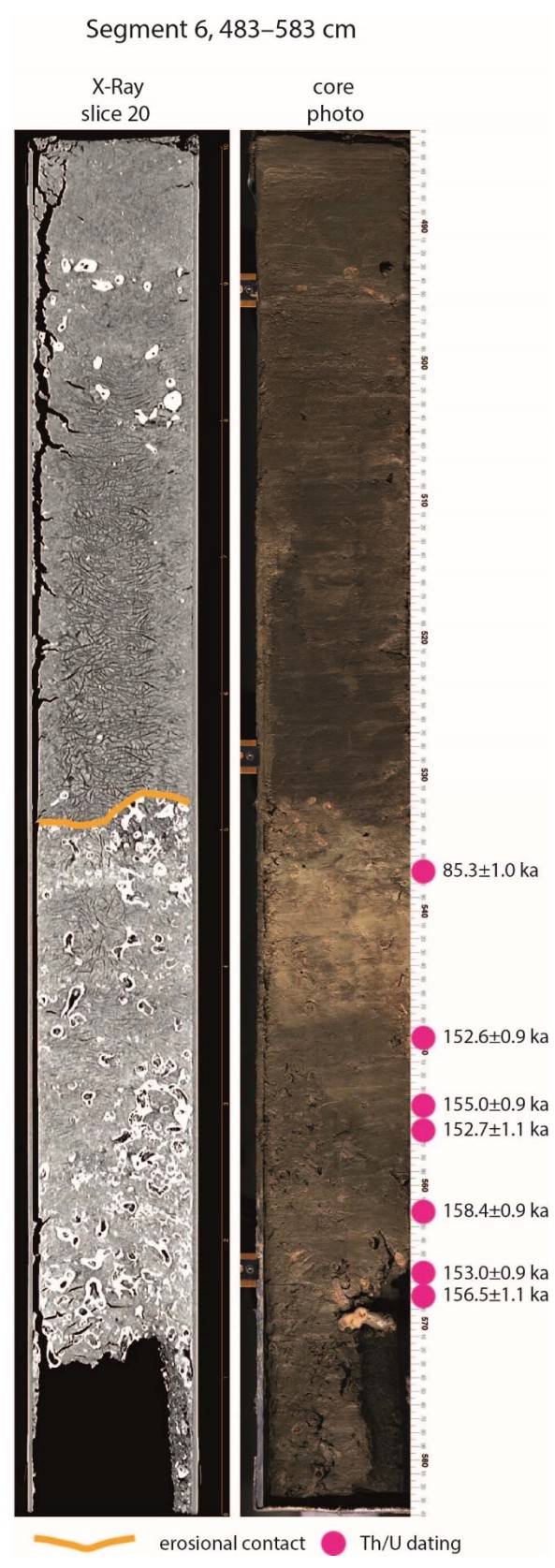

**Appendix Figure F:** CT image (left) and photography (right) of segment 383-483 cm of core M125-34-2. Erosional contacts are marked by thick black lines, positions of Th/U datings with calibrated age are indicated by magenta dots.

## Data availability

All data presented in this study will be made available in the Pangaea data base ([www.pangaea.de](www.pangaea.de)).


## Author contribution

AB and JR designed the study. AB, MD, JT, GA, DN, AK, and JR were involved in sampling and data generation. All authors contributed to data interpretation and manuscript writing.


## Competing Interests

The authors declare that they have no conflict of interest.

## Acknowledgements:

We thank Captain and crew of R/W Meteor for their support during Expedition M125. We further kindly acknowledge I. Glass (X-Ray diffractometry), D. Bergmann-Dörr (carbon and nitrogen analysis), S. Brzelinski, C. Catunda, S. Fessler, J. Fiebig, S. Hofmann, B. Knape, (stable oxygen and carbon isotope analysis), N. Gehre (sample preparation and foraminiferal selection), L. Frede (sortable silt analysis), for support during sample preparation and analyses. The authors are grateful to W. Stiller and S. Skornitzke from the Heidelberg University Hospital for performing the CT 600 scanning. AB was funded by the Deutsche Forschungsgemeinschaft (DFG project HO5927/1-1). JT received funds from the MARUM Cluster of Excellence 'The Ocean Floor – Earth's Uncharted Interface' (Germany´s Excellence Strategy – EXC-2077 – 390741603 of the Deutsche Forschungsgemeinschaft DFG).

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
