# Peer review of "Monsoonal forcing of cold-water coral growth off south-eastern Brazil during the past 160 kyrs"

_Biogeosciences, 2020_

## Referee Comment (RC1) · Robin Fentimen (Referee) · 24 Aug 2020

Title: "Monsoonal forcing controlled cold water coral growth off southeastern Brazil during the past 160 kyrs"

General comments

Based on a multiproxy study of sediment cores M125-34-2 (on-mound) and M125-50-3 (off-mound), Bahr et al. set out to constrain the long-term development of Bowie Mound and to understand the environmental forcing behind its formation. They essentially conclude that an enhanced delivery of terrestrial organic matter during Heinrich Stadials (HS 1, 4 and 6) played an important role on cold-water coral growth off SE Brazil. They compare their set of sedimentological and geochemical proxies to previously published data in the area. As such, the study is based on a solid and plentiful number of proxies. The chronostratigraphy of the core is well constrained and allows, in my opinion, for a satisfying interpretation of the data. The discussion is to the point and not too lengthy, it could even go a bit more in depth (see Comment #3). The stable isotope analyses of infaunal foraminiferal tests is where the most improvement could be done. This is detailed in Specific comment #1. The conclusions drawn by the authors are arguably by the choice of method, and may have yielded more precise results and details if the approach would have been different (and more taxonomically precise; see comment #1).

The quality of the English in the manuscript is at times insufficient. Some corrections are listed in the section "Technical corrections". In addition to these, the manuscript would need a few extra proof readings to reach the desired quality. I am however confident that this can be done shortly and satisfyingly by the authors.

All things considered, I would be happy to recommend this manuscript for publication in Biogeosciences if the points below (plus the English in the text) are addressed by the authors. The manuscript presents a novel and interesting dataset that falls within the scope of the journal and will be of interest to its readers.

Specific comments

Comment #1: My main comment concerns the grouping of different Uvigerina species for stable isotope analyses. The authors mention the genus Uvigerina spp in the material and methods. It would be good to mention here the species considered in this grouping. How many species were considered in the grouping? Was one species more abundant? Is one species more abundant during specific intervals (e.g. within CWC-bearing intervals)? Indeed, it has been demonstrated that the response to trophic

conditions is species-specific for the genus Uvigerina (see for example, Theodor et al., 2016 Marine Micropaleontology). Uvigerina mediterranea is for example better suited than U. peregrina to reconstruct trophic conditions, since it is more of an opportunistic species. Uvigerinids do not share the same ecological preference (see for example Fontanier et al., 2006), thus I am quite skeptical about this grouping. In my opinion, the grouping of Uvigerinids together weakens the use of stable isotope analyses performed on their tests, since it is not monospecific (as mentioned by the authors at Line 339). Hence the conclusions of section 5.1.1 are not as solid as they could be if authors considered species alone. Although I understand that this approach was chosen as a second choise because of the lack of material, I suggest that the authors should address more and discuss this point more in detail in the material and methods section and in Section 5.1.1.

I recommend plotting the $\delta$13C of individual Uvigerina species and then to compare this to the results of the grouping (all species combined). The scatter of the normalized data may possibly be due to the effect of the grouping. This can be easily verified by isolating different Uvigerina species and adding an extra colour code to Fig. 4A. As such, the results presented by the authors would be clearer.

Comment #2: Although the interpretations and conclusions are in my opinion sound, the association of coral proliferation with HS 4 does not seem as clear as for HS 1 and 6. There is an offset between the Ti/Ca and speleotherm records presented in Figure 5 with the coral proliferation phase. Is this due to an age model uncertainty? I think this offset should be discussed a bit more in detail.

Comment #3: It would be appreciated if the authors took the discussion one step further by comparing the environmental forcing observed in the study area to other CWC settings, e.g. along the East Atlantic margin or in the Mediterranean. This could be done in the last section of the discussion. For example, Wienberg et al. (2010) suggested that aeolian dust had a local fertilization effect on coral growth in the Gulf of Cadiz, whilst Fentimen et al. (2020) propose that fluvial input triggered coral proliferation during Greenland Interstadial 1 in the Western Mediterranean (Melilla Mound Province). Authors should also consider the work of Mienis et al. in the Western Atlantic. As such, the conclusions of the authors fit in with other previous observations and add new evidence. This is something that I believe should be better highlighted and deserves to be developed. The last statement of the conclusion that "This study (. . .) points at a hitherto unrecognized intimate coupling between continental hydroclimate and ecological changes in the deep ocean" is in this sense too bold and should be tempered. Indeed, previous studies already suggest this.

Also the link between coral growth and monsoonal forcing is only written and stated clearly in the title. No mention of the term "monsoonal forcing" is done in the discussion and conclusion. I think that if the title uses this term, it should also clearly be stated and discussed in the discussion (noticeably in section 5.2).

Technical corrections Title: "cold-water coral", missing "-" Line 25: "located at" and not "located in" Line 42: "constrained" and not "constraint" Lines 48 to 52: These two sentences need to be rephrased; I cannot get the meaning of the sentences as they are. Especially in the second sentence, the verb is missing ("Changes the species (. . .)"). Line 53: Explain the abbreviations POC and DOC the first time you introduce them, some readers may not be acquainted with these. Line 55: This sentence needs to be reworked, it is not understandable as it is: "Note, however, that similar studies in the feeding in the properties (. . .)". Do the authors mean feeding properties / feeding behaviour? Line 58: In the sentence: "All affect the capacity (. . .)" I would suggest repeating the word parameters or variables, i.e. "All these parameters (or environmental variables) affect the capacity (. . .)". Line 61: check the grammar: "to play a role in" not "to play a role for" Lines 62 to 64: The end of this sentence is not clear, consider reworking it. For example: "(. . .) importance of surface productivity in providing food to the deep ocean". Line 70: I would suggest not to start the sentence with an abbreviation (Here CWC). Line 72: "Adapted" and not "adopted" Line 82: rephrase the sentence: "demonstrates for the first time" instead of "for the first time demonstrates".

Lines 81 to 83: The combined use in this sentence of "for the first time" and "a so far underestimated" is possibly a bit redundant. I would recommend less emphasizing in this sentence. There is no need to say it is "so far underestimated" if it is the first time it has been observed. Figure 1: Numbers on the hydrographic section (top left) are barely readable. I would suggest increasing the size of these. Line 133: Spelling: "half" not "halve" Line 137: Correct the beginning of the sentence: "Core M125-34-2" instead of "The core (. . .)" Line 145: Correct the beginning of the sentence: "To constrain" instead of "for constraining" Line 146: Correct the English: "was sampled at (or sampled every 10 cm)", instead of "was sampled in" Line 168: "Half" instead of "halve" Line 181: "at Heidelberg University" instead of "at the Heidelberg University", or rephrase: "at the Department of Geosciences, Heidelberg University". Line 184: "were analysed with the Diffract Suite (. . .)" instead of "was analysed with Diffract Suite (. . .)" Line 185: Avoid using the passive form to often when possible. For example here, rather write: "The Rietveld refinement program DIFFRAC.TOPAS (Bruker Software) was used to perform quantitative phase analysis". Line 195: "Weighed" instead of "weighted". The verb is "to weigh" (thus weighed in the past tense), the noun is "weight". Line 195: I would suggest rather writing "filled to the top" instead of "filled until capacity". Line 198: correct: "(. . .) and put into an ultrasonic bath", instead of "(. . .), put into an (. . .)" Lines 204 and 205: Is there a mistake here: "The high number of replicates resulted from". Do you mean: "resulted in" ? Line 257 and 258: No capital letter given to "core" (write "core") Figure 3: The symbol (white diamond) of Uvigerina spp. appears to be missing on the figure. Line 369: correct to "seemed to have" Line 382 to 384: Check the sentence for grammar: "increasing" instead of "increase", "suggests" instead of "suggest". Line 483: The sentence needs to be rephrased, it reads: "Due to their baffling capacity, the additional sedimentary input would have aided mound formation". I would recommend rather writing: "Due to the baffling capacity of CWCs, the (. . .)". As it is, the sentence suggests that the mound baffles sediment, whilst it is the corals not the mound in itself.

---

## Referee Comment (RC2) · Anonymous Referee #2 · 2 Sep 2020

Comments to the Author The manuscript submitted here investigates the impact of monsoonal variability on CWC growth in last 160 Kyrs. While the authors present a manuscript with compelling arguments; that is likely to be of interest to readers of Biogeosciences, I have a few of concerns that should be addressed before publication.

1. Authors try to show how the monsoon impacted CWC growth without providing any direct correlation between the two, which is a simple statistical analysis to do. 2. The discussion section needs to be streamlined towards the main objective of the manuscript, which now rather seems to be a collection of different points without the central theme. It's difficult for a reader to go through the whole discussion and find

exactly where the authors prove their central claim. While discussing many proxies is necessary for a paper like this, it's also important to stress how these proxies help to prove your central claim, which is something lacking in the manuscript. 3. While the growth of CWC during HS events is very evident visually, why the CWC growth was not observed during MIS3 and MIS4 is still not clearly explained. While TOC is the only proxy that was different during these stages but high TOC didn't promote CWC growth at 20-40m depth. So it seems that TOC is not a singular factor affecting CWC growth. While authors have explained water currents and terrigenous input as some other proxies to impact CWC growth, they seem to be fluctuating a lot in all the MIS stages and hence fail to shed any light on what stopped CWC from growing during MIS3 and MIS4. 4. The figures captions throughout the manuscript describe what is shown in the figure, but don't provide the reader with any additional information such as calling attention to the significant result. The message shown by the figure is left entirely up to the reader to decipher. Moreover, in some figures authors have added depth and in some age. It would be best if authors add age and depth in all the figures. 5. Line 48: "The most common framework-forming CWC comprise. "This sentence doesn't make sense. It is either incomplete or needs to be restructured. 6. The next sentence in line 49 "Changes the species...." Is also incomplete and hence doesn't provide context. 7. Line 55: "in" repeated "similar studies in the feeding in the properties..." 8. Line 72: It should be "adapted" instead of "adopted". 9. Line 77 -79: "This setting allows us..... growth at Bowie mound". It is a repetition of what has been already said in previous sentences. 10. Line 369: "AAIW seemed to had an insignificant" It should be "to have had" or "AAIW had" depending on what authors want to say exactly. 11. Line 384: "does not necessarily led to". It should be "lead"

---

## Author Comment (AC1) · 20 Sep 2020

Response to Robin Fentimen (Referee 1) First we would like to thank Robin Fentimen for his careful and supportive evaluation of our manuscript. Below we provide a point-to-point response ("R") to the original comments ("C").

[C] Bahr et al. set out to constrain the long-term development of Bowie Mound and to understand the environmental forcing behind its formation. They essentially conclude that an enhanced delivery of terrestrial organic matter during Heinrich Stadials (HS 1, 4 and 6) played an important role on cold-water coral growth off SE Brazil. They compare their set of sedimentological and geochemical proxies to previously published data in

the area. As such, the study is based on a solid and plentiful number of proxies. The chronostratigraphy of the core is well constrained and allows, in my opinion, for a satisfying interpretation of the data. The discussion is to the point and not too lengthy, it could even go a bit more in depth (see Comment #3). The stable isotope analyses of infaunal foraminiferal tests is where the most improvement could be done. This is detailed in Specific comment #1. The conclusions drawn by the authors are arguably by the choice of method, and may have yielded more precise results and details if the approach would have been different (and more taxonomically precise; see comment #1).

[R] As detailed below we provide more details on the stable isotope analysis. Although we are aware of the potential limitations of this method (especially of using endobenthic d13C for bottom water reconstructions due to the absence of sufficient amounts of epibenthic foraminifera) we are confident that the results provide a robust assessment of major shifts in the bottom water regime. However, we agree with the reviewer that an investigation of the benthic foraminiferal communities might provide useful insights into the changes of trophic levels and nutrient supply to the continental margin (such as done in Fentimen et al., 2020). This is clearly beyond the scope of this study but might well be tackled as part of a follow-up research.

[C] The quality of the English in the manuscript is at times insufficient. Some corrections are listed in the section "Technical corrections". In addition to these, the manuscript would need a few extra proof readings to reach the desired quality. I am however confident that this can be done shortly and satisfyingly by the authors.

[R] We thank both Reviewers for pointing at typos and incorrect language. They were all corrected and we also had the text proof-read by a native speaker to improve the quality of the English. Please note that we also changed the title into "Monsoonal forcing of cold-water coral growth off south-eastern Brazil during the past 160 kyrs" instead of "Monsoonal forcing controlled cold water coral growth...".

[C] All things considered, I would be happy to recommend this manuscript for publication in Biogeosciences if the points below (plus the English in the text) are addressed by the authors. The manuscript presents a novel and interesting dataset that falls within the scope of the journal and will be of interest to its readers.

[R] We thank R. Fentimen for his positive assessment of our study and hope that we could sufficiently address the concerns raised in the specific comments below.

[C] Specific comments Comment #1: My main comment concerns the grouping of different Uvigerina species for stable isotope analyses. The authors mention the genus Uvigerina spp in the material and methods. It would be good to mention here the species considered in this grouping. How many species were considered in the grouping?

[R] We combined U. peregrina, and U. proboscidae, as now also stated in Section 3.7.1: "For each sample 1–3 tests of Uvigerina spp. (U. peregrina and U. proboscidea)...".

[C] Was one species more abundant? Is one species more abundant during specific intervals (e.g. within CWC bearing intervals)?

[R] The dominant species is U. peregrina constituting roughly 75 % of all species of the genus Uvigerina in the samples (which rarely contained more than a total of 3 Uvigerina specimens). Monospecific selection of Uvigerina peregrina would have led to significantly more gaps in the stable isotope record. To obtain enough material we therefore mostly used 3 specimens of Uvigerina, when necessary pooling different species for one analysis. We didn't, however, denote the number of individual tests selected from each species per sample. Based on previous inspection of the samples, there is no notable turnover in the relative proportion of the respective species throughout the core, hence, we do not expect a bias in the d13C record due to shifts in the relative abundances of those species.

[C] Indeed, it has been demonstrated that the response to trophic conditions is speciesspecific for the genus Uvigerina (see for example, Theodor et al., 2016 Marine Micropaleontology). Uvigerina mediterranea is for example better suited than U. peregrina to reconstruct trophic conditions, since it is more of an opportunistic species. Uvigerinids do not share the same ecological preference (see for example Fontanier et al., 2006), thus I am quite skeptical about this grouping. In my opinion, the grouping of Uvigerinids together weakens the use of stable isotope analyses performed on their tests, since it is not monospecific (as mentioned by the authors at Line 339).

[R] We agree with the reviewer that single-species selection would be the preferred choice for the construction of any stable isotope record, however, we opted for obtaining a record as complete as possible which limited us to combining different species as well as different genera (Uvigerina and Planulina). As mentioned before, a more in-depth study of faunal turnovers in the benthic foraminiferal communities including stable carbon isotope data from different species with different ecological preferences might be a highly valuable follow-up project but is outside the scope of this study. We would like to point out that when discussing bottom-water variability as a potential driver of CWC growth we focus on the interpretation of the signal of epibenthic P. wuellerstorfi (as stated in the original manuscript text), which provides a more robust insight into bottom-water variability than infaunal species like Uvigerina. The higher scatter evident in the Uvigerina spp. d13C data compared to the data obtained on P. wuellerstorfi is likely due to the influence of changes in the isotopic composition of pore-water DIC but might also derive from the pooling of different species. However, in the case of U. peregrina and U. proboscidae Rathburn et al. (1996) found an offset of only 0.1–0.2 ‰. This is much smaller than the 0.8–1.4‰ offset between U. peregrina and U. mediterranea published by Theodor et al. (2016), indicating that U. peregrina and U. proboscidae occupy a similar habitat. We nevertheless now state the caveats arising from pooling the different Uvigerina species in the manuscript (section 5.1.1): "The resulting normalized data exhibits a considerable scatter in the Uvigerina spp. record due to the pooling of different species, and hence only allows for a discussion of major shifts in the isotopic composition. However, the d13C data of Uvigerina spp. and P.

[Figure]

wuellerstorfi do not provide compelling evidence for distinctly depleted values during phases of CWC growth compared to CWC-barren intervals (Fig. 4). Although d13CUvi might be influenced by isotopic variations of dissolved inorganic carbon of pore-water (Zahn et al., 1986) and inter-species offsets (Rathburn et al. 1996; Theodor et al., 2016), we nevertheless consider it as appropriate for reconstructing major changes in the bottom-water signature."

[C] Hence the conclusions of section 5.1.1 are not as solid as they could be if authors considered species alone. Although I understand that this approach was chosen as a second choise because of the lack of material, I suggest that the authors should address more and discuss this point more in detail in the material and methods section and in Section 5.1.1.

[R] As suggested, we added further details on the selection of different Uvigerina species in the method section and provide a discussion about the caveats in Section 5.1.1 (see comments above).

[C] I recommend plotting the d13C of individual Uvigerina species and then to compare this to the results of the grouping (all species combined). The scatter of the normalized data may possibly be due to the effect of the grouping. This can be easily verified by isolating different Uvigerina species and adding an extra colour code to Fig. 4A. As such, the results presented by the authors would be clearer.

[R] Unfortunately, we did not denote which species we selected for each analysis; in some samples we also had to lump different species of Uvigerina into one sample to accumulate enough material. We agree with the reviewer that the grouping might be at least partly responsible for the relatively large scatter of the normalized d13C data of Uvigerina spp.; a respective remark has been added to the discussion in Section 5.1.1 (see above). As stated above, we nevertheless consider our results obtained from the d13C analyses to serve its purpose (i.e., tracing bottom-water variability), as we are discussing only major fluctuations in the d13C signal (not the scatter) and primarily

rely on the results of single-species epibenthic P. wuellerstorfi, which is naturally a better tracer of bottom-water variability than any endobenthic species.

[C] Comment #2: Although the interpretations and conclusions are in my opinion sound, the association of coral proliferation with HS 4 does not seem as clear as for HS 1 and 6. There is an offset between the Ti/Ca and speleotherm records presented in Figure 5 with the coral proliferation phase. Is this due to an age model uncertainty? I think this offset should be discussed a bit more in detail.

[R] We argue that the temporal offsets of CWC proliferation phases vs. monsoonal indicators (i.e. the Ti/Ca record and travertine/speleothem growth phases) during HS 2-5 is largely due to age model uncertainties, as the age model of the marine Ti/Ca record is based on AMS 14C dating and benthic isotope stratigraphy, introducing an error of ~2 kyrs for this interval (Campos et al., 2019). It has also to be considered that HS 1 and 6 go along with humid phases that are longer and more pronounced than HS 2-5. Hence, the expected CWC response to the increased run-off should be less pronounced during HS 2-5.

We added these aspects to the discussion of the presumed monsoonal forcing of the CWC signal in Section 5.2: "...The most distinct CWC proliferation phases in fact took place during phases of anomalously strong monsoonal precipitation during the pronounced Heinrich Stadials (HS) 1 and 6 as well as (within age model uncertainties) also during the shorter and less severe humid phases corresponding to HS 2–5 (Fig. 5F, G)."

[C] Comment #3: It would be appreciated if the authors took the discussion one step further by comparing the environmental forcing observed in the study area to other CWC settings, e.g. along the East Atlantic margin or in the Mediterranean. This could be done in the last section of the discussion. For example, Wienberg et al. (2010) suggested that aeolian dust had a local fertilization effect on coral growth in the Gulf of Cadiz, whilst Fentimen et al. (2020) propose that fluvial input triggered coral prolif-

eration during Greenland Interstadial 1 in the Western Mediterranean (Melilla Mound Province). Authors should also consider the work of Mienis et al. in the Western Atlantic.

[R] We value the suggestion to broaden the discussion and now include the reference to other studies that also propose links between terrestrial input and CWC proliferation. These studies include the suggested papers by Wienberg et al. (2010) and Fentimen et al. (2020; Frontiers in Marine Science). We also reference Hanz et al. (2019) who inferred that terrestrial dissolved nutrients aided CWC growth off Angola. We added a respective statement at the end of the first paragraph of Section 5.2: "The here proposed link between Monsoonal activity and CWC growth is in line with studies from the western Mediterranean Sea (Fentimen et al., 2020), the Gulf of Cadiz (Wienberg et al., 2010) and the tropical eastern Atlantic off Angola (Hanz et al., 2019) which inferred that terrestrial input via dust or fluvial run-off ultimately fueled thriving CWC colonies." The mentioned publications of Mienis et al. only referred rather indirectly to terrigenous sediment input as a driver of CWC proliferation, thus we omitted reference to these publications in this context.

[C] As such, the conclusions of the authors fit in with other previous observations and add new evidence. This is something that I believe should be better highlighted and deserves to be developed. The last statement of the conclusion that "This study (. . .) points at a hitherto unrecognized intimate coupling between continental hydroclimate and ecological changes in the deep ocean" is in this sense too bold and should be tempered. Indeed, previous studies already suggest this.

[R] As suggested we toned down the respective sentence to "This study thus presents a prime example of the intimate coupling between continental hydroclimate and ecological changes in the deep ocean." In the same sense we modified the last sentence in the abstract to "Our study thus emphasizes the impact of continental climate variability on a highly vulnerable deep-marine ecosystem".

[C] Also the link between coral growth and monsoonal forcing is only written and stated clearly in the title. No mention of the term "monsoonal forcing" is done in the discussion and conclusion. I think that if the title uses this term, it should also clearly be stated and discussed in the discussion (noticeably in section 5.2).

[R] We now state more explicitly the connection of monsoonal forcing on CWC growth in the discussion (see also response to comment #2) and in the Conclusions ("We find that intervals of high CWC abundance are primarily related to . . . enhanced monsoonal precipitation in eastern Brazil."). In the abstract we now phrase: "Our results indicate a multi-factorial control on CWC growth and mound formation at Bowie Mound during the past ∼160 kyrs, which reveals distinct formation pulses during glacial high northern latitude cold events (Heinrich Stadials, HS) largely associated with anomalously strong monsoonal rainfall over the continent."

[C] Technical corrections Title: "cold-water coral", missing "-"

[R] Corrected.

[C] Line 25: "located at" and not "located in"

[R] Corrected.

[C] Line 42: "constrained" and not "constraint"

[R] Corrected.

[C] Lines 48 to 52: These two sentences need to be rephrased; I cannot get the meaning of the sentences as they are. Especially in the second sentence, the verb is missing ("Changes the species (. . .)").

[R] We thank the reviewer for pointing at this typographic mishap. These two sentences now read "The most common framework-forming CWC comprise of Lophelia pertusa (recently assigned to the genus Desmophyllum by Addamo et al., 2016), Macropora oculata, Solenosmilia variabilis, Bathelia candida, and Enallopsammia profunda (e.g.,

Mangini et al., 2010; Frank et al., 2011; Muñoz et al., 2012; Hebbeln et al., 2014; Raddatz et al., 2020)."

[C] Line 53: Explain the abbreviations POC and DOC the first time you introduce them, some readers may not be acquainted with these.

[R] Done.

[C] Line 55: This sentence needs to be reworked, it is not understandable as it is: "Note, however, that similar studies in the feeding in the properties (. . .)". Do the authors mean feeding properties / feeding behaviour?

[R] This sentence and the following now reads "However, we note that similar studies on the feeding preferences of S. variabilis, the dominant framework-building CWC at the herein investigated Bowie Mound (Raddatz et al. 2020) are still missing. Additionally, changes in the properties and spatial configuration of ambient intermediate- or deep-water masses may also strongly impact CWC through changes in the dissolved oxygen concentration and the seawater parameters pH, alkalinity and carbonate-ion concentration."

[C] Line 58: In the sentence: "All affect the capacity (. . .)" I would suggest repeating the word parameters or variables, i.e. "All these parameters (or environmental variables) affect the capacity (. . .)".

[R] The sentence has been revised to "All these parameters affect. . .".

[C] Line 61: check the grammar: "to play a role in" not "to play a role for"

[R] Corrected.

[C] Lines 62 to 64: The end of this sentence is not clear, consider reworking it. For example: "(. . .) importance of surface productivity in providing food to the deep ocean".

[R] We rephrased the sentence according to the suggestion of the reviewer.

[C] Line 70: I would suggest not to start the sentence with an abbreviation (Here CWC).

[R] We changed the start of the sentence to "The presence of CWC-bearing mounds. . .".

[C] Line 72: "Adapted" and not "adopted"

[R] Corrected.

[C] Line 82: rephrase the sentence: "demonstrates for the first time" instead of "for the first time demonstrates". C4

[R] Corrected.

[C] Lines 81 to 83: The combined use in this sentence of "for the first time" and "a so far underestimated" is possibly a bit redundant. I would recommend less emphasizing in this sentence. There is no need to say it is "so far underestimated" if it is the first time it has been observed.

[R] We deleted "for the first time" to avoid redundancy.

[C] Figure 1: Numbers on the hydrographic section (top left) are barely readable. I would suggest increasing the size of these.

[R] Done.

[C] Line 133: Spelling: "half" not "halve"

[R] Corrected.

[C] Line 137: Correct the beginning of the sentence: "Core M125-34-2" instead of "The core (. . .)"

[R] Corrected.

[C] Line 145: Correct the beginning of the sentence: "To constrain" instead of "for constraining"

[R] Corrected.

[C] Line 146: Correct the English: "was sampled at (or sampled every 10 cm)", instead of "was sampled in"

[R] Corrected.

[C] Line 168: "Half" instead of "halve"

[R] Corrected.

[C] Line 181: "at Heidelberg University" instead of "at the Heidelberg University", or rephrase: "at the Department of Geosciences, Heidelberg University".

[R] Corrected.

[C] Line 184: "were analysed with the Diffract Suite (. . .)" instead of "was analysed with Diffract Suite (. . .)"

[R] Corrected.

[C] Line 185: Avoid using the passive form to often when possible. For example here, rather write: "The Rietveld refinement program DIFFRAC.TOPAS (Bruker Software) was used to perform quantitative phase analysis".

[R] Corrected as suggested.

[C] Line 195: "Weighed" instead of "weighted". The verb is "to weigh" (thus weighed in the past tense), the noun is "weight".

[R] Corrected.

[C] Line 195: I would suggest rather writing "filled to the top" instead of "filled until capacity".

[R] Corrected.

[C] Line 198: correct: "(. . .) and put into an ultrasonic bath", instead of "(. . .), put into

an (. . .)"

[R] Corrected.

[C] Lines 204 and 205: Is there a mistake here: "The high number of replicates resulted from". Do you mean: "resulted in" ?

[R] We changed the sentence into "The relatively large inter-sample variability that is likely caused by. . .".

[C] Line 257 and 258: No capital letter given to "core" (write "core")

[R] Corrected.

[C] Figure 3: The symbol (white diamond) of Uvigerina spp. appears to be missing on the figure.

[R] We added the diamond (symbol for P. wuellerstorfi) to the legend.

[C] Line 369: correct to "seemed to have"

[R] Corrected.

[C] Line 382 to 384: Check the sentence for grammar: "increasing" instead of "increase", "suggests" instead of "suggest".

[R] We did not apply these suggested changes, as "increasing" does not fit into the sentence's structure and "data", the reference of "suggest" is in plural.

[C] Line 483: The sentence needs to be rephrased, it reads: "Due to their baffling capacity, the additional sedimentary input would have aided mound formation". I would recommend rather writing: "Due to the baffling capacity of CWCs, the (. . .)". As it is, the sentence suggests that the mound baffles sediment, whilst it is the corals not the mound in itself.

[R] We thank the reviewer for pointing out this ambiguity. We rephrased the sentence as suggested.

---

## Author Comment (AC2) · 20 Sep 2020

Response to Referee 2 First we would like to acknowledge Reviewer 2 for his/her constructive evaluation of our manuscript. Below we provide a point-to-point response ("R") to the original comments ("C").

[C] The manuscript submitted here investigates the impact of monsoonal variability on CWC growth in last 160 Kyrs. While the authors present a manuscript with compelling arguments; that is likely to be of interest to readers of Biogeosciences, I have a few of concerns that should be addressed before publication. 1. Authors try to show how the monsoon impacted CWC growth without providing any direct correlation between the

two, which is a simple statistical analysis to do. [R] We thank Reviewer 2 for the suggestion of performing a statistical test to validate the proposed monsoonal impact on the CWC growth. For this purpose we computed the linear correlation between CWC occurrences (as shown in Figure 5H) and the Ti/Ca ratio of Core M125-95-3 using the Monte-Carlo-based SurrogateCor function implemented in the "astrochron" package in R (Meyers, 2014), which has been specifically designed for correlation of time series with a different temporal resolution. The resultant correlation coefficient r = 0.56 (p=0.01) corroborates a significant correlation even considering potential mismatches due to age model uncertainties and non-linear proxy behavior. We include these new statistical results in the caption of Fig. 5: "Note the good match between CWC occurrences and enhanced monsoonal activity on the continent (correlation between Ti/Ca and CWC frequency: r = 0.56, p = 0.02; computed using the SurrogateCor function of the R-package "astrochron"; Meyers, 2014)". Notably, these results also support the already stated results of a discriminant analysis which showed that proxies reflecting terrigenous run-off (Corg/Ntotal) and weathering (albite/kaolinite) are good predictors for CWC occurrences (cf. end of Section 5.1.3 "Influence of the continental hydrological cycle").

[C] 2. The discussion section needs to be streamlined towards the main objective of the manuscript, which now rather seems to be a collection of different points without the central theme. It's difficult for a reader to go through the whole discussion and find exactly where the authors prove their central claim. While discussing many proxies is necessary for a paper like this, it's also important to stress how these proxies help to prove your central claim, which is something lacking in the manuscript.

[R] We recognize the concern by Reviewer 2 (partly also raised by Reviewer 1), and put more emphasis on the main message of the paper, i.e. the direct link between continental hydroclimate and CWC growth at the continental slope. We now stress this link right at the onset of Section 5 "Results and Discussion": "…We argue that the most dominant environmental factor for triggering CWC growth was elevated river runoff during periods of strong monsoonal rainfall in the coastal hinterland, which provided nutrients and organic matter that enhanced the food supply of CWC colonies." We further emphasize the role of monsoonal rainfall as a decisive factor for CWC growth as also requested by Reviewer 1 (this affects Abstract, Section 5.2, and Conclusions).

[C] 3. While the growth of CWC during HS events is very evident visually, why the CWC growth was not observed during MIS3 and MIS4 is still not clearly explained. While TOC is the only proxy that was different during these stages but high TOC didn't promote CWC growth at 20-40m [REMARK: this likely refers to 20-40 cm] depth. So it seems that TOC is not a singular factor affecting CWC growth. While authors have explained water currents and terrigenous input as some other proxies to impact CWC growth, they seem to be fluctuating a lot in all the MIS stages and hence fail to shed any light on what stopped CWC from growing during MIS3 and MIS4.

[R] The Reviewer refers here to the CWC barren interval between 20–40 cm core depth. We note that this interval is characterized by intermediate TOC contents that are not as high as during the main CWC growth phases between 70-180 cm and below 530 c (Fig. 6A). As already stated in the original text (see section 5.1.2: "However, as suggested in the previous section, there are multiple factors necessary for stimulating coral growth at Bowie Mound"), we assume a multi-factorial control of CWC proliferation phases. Organic matter supply likely played a major role but other factors such as hydrodynamic conditions might have interfered as well. In this line the absence of CWC during large portions of MIS 4 could be well explained by the generally low TOC contents that point to the lack of organic matter supply inhibiting CWC growth. We now state this connection explicitly in Section 5.1.2: "...while the long CWC-barren interval between 200–460 cm is characterized by relatively low TOC contents...". According to the stratigraphic assignment this specific interval encompasses early MIS 4, not MIS 3; to make this distinction clearer we modified the age assignments in Figures 4 and 6 and modified the respective paragraph in Section 4: "The section between EII and EIII on the other hand has relatively uniform d18O values around 4.3 ‰ (d18OUvi) and 3.4 ‰

(d18OPlan), respectively, which matches M125-50-3 d18OUvi values during MIS 4 to 2. A Th/U date at the top of this section at 117 cm reveals an age of 34 ka, while CWC ages from slightly deeper in the core (between 131-190 cm) fall within the range of 60–63 ka (MIS 4). As d18OUvi values between EII and EIII are less depleted than MIS 5 samples of reference site M125-50-3, we infer that those sediments were most likely deposited during MIS 4 and did not reach into MIS 5. Hence, it hence appears that deposits of MIS 2 and large parts of MIS 3 are not present in core M125-34-2, either due to non-deposition or subsequent erosion (note the prominent erosive surface EII). This age assignment would also imply that the extended CWC-free portion from 200 to 465 cm was deposited within a short period of approximately 8 kyr during MIS 4 (62.2 ka as the oldest Th/U dates and ∼70 ka as the MIS 4/5 boundary)."

[C] 4. The figures captions throughout the manuscript describe what is shown in the figure, but don't provide the reader with any additional information such as calling attention to the significant result. The message shown by the figure is left entirely up to the reader to decipher.

[R] We are aware that the figures and the associated discussion of the factors influencing CWC growth phases are quite complex. We therefore acknowledge the suggestion by Reviewer 2 to provide more information on the interpretation of a Figure's content in the respective caption. We hence added the following sentences to the captions: Fig. 4: "Phases of CWC proliferation appear to require background state of high hydrodynamics conditions (elevated ln(Zr/Al) and (SS) Ȉ) but do not show an influence of deep-water mass variability (d13C).." Fig. 5: "Note the good match between CWC occurrences and enhanced monsoonal activity on the continent (correlation between ln(Ti/Ca) and CWC frequency: r = 0.56, p = 0.02; computed using the SurrogateCor function of the R-package "astrochron"; Meyers, 2014)." Fig. 6: "Note that high CWC abundances fall into intervals of high TOC and increased weathering due to an intensified continental hydrological cycle."

[C] Moreover, in some figures authors have added depth and in some age. It would be

best if authors add age and depth in all the figures.

[R] We would like to note that we already provide depth and age scales on Figures, when appropriate (Figs. 3, 4, 6). For Figure 5 we followed the reviewer's suggestion and added the depths intervals denoting the phases of CWC proliferation at Bowie Mound on top of the figure; a respective comment has been added to the caption ("Red bars indicate periods of enhanced CWC growth at Bowie Mound, with the respective depths in Core M125-34-2 annotated").

[C] 5. Line 48: "The most common framework-forming CWC comprise. "This sentence doesn't make sense. It is either incomplete or needs to be restructured.

[R] This sentence has been rephrased (see also reply to respective comment by Reviewer 1).

[C] 6. The next sentence in line 49 "Changes the species. . .." Is also incomplete and hence doesn't provide context.

[R] This sentence has been rephrased (see also reply to respective comment by Reviewer 1).

[C] 7. Line 55: "in" repeated "similar studies in the feeding in the properties. . ."

[R] We rephrased the sentence to "... similar studies on the feeding preferences of S. variabilis, the dominant framework-building CWC discussed in this study, are still pending".

[C] 8. Line 72: It should be "adapted" instead of "adopted".

[R] Corrected.

[C] 9. Line 77 -79: "This setting allows us. . ... growth at Bowie mound". It is a repetition of what has been already said in previous sentences.

[R] This sentence has been removed (see also reply to respective comment by Reviewer 1).

[C] 10. Line 369: "AAIW seemed to had an insignificant" It should be "to have had" or "AAIW had" depending on what authors want to say exactly.

[R] Corrected to "seemed to have".

[C] 11. Line 384: "does not necessarily led to". It should be "lead"

[R] Corrected.

REFERENCE Meyers, S., 2014. Astrochron: An R package for astrochronology. Available at http: org/web/packages/astrochron/index.html.

---

## Author Response (AR1)

Dear Editor,
First, we would like to acknowledge your careful handling of the manuscript and the suggestions you made in your comments. Please find below our response (in black) to each point raised by you (in blue).

A. I would suggest a sedimentation rate profile to further strengthen your argument.

We agree that displaying the sedimentation rate helps to foster our arguments. We hence added sedimentations rates obtained from the Th/U-dated CWC-bearing intervals to Fig. 3, also indicating the indirectly inferred sedimentation rate for the CWC-barren interval between 200-465 cm. The figure caption has been modified accordingly.

B. Do you think a high input of nutrients during glacials will enhance the deposition of greater marine-derived organic matter? However, that signal is not strong enough!!

As we infer in the discussion the input of nutrients from run-off fertilizes the surface ocean and will likely boost surface productivity. However, the C/N ratio still indicates that most organic matter in the CWC bearing intervals derives from land, clearly dominating over marine organic matter. Hence, the potentially increased marine productivity appears secondary to the terrigenous organic matter input when it comes to fueling CWC growth.

C. The delta 13C (TOC) data does not show a sharp drop except at MIS 6!! Although it was expected during LGM due to the spreading of grassland and aridity. We see that routinely at many places. These aspects need some explanation.

We agree that MIS 6 should show a strong trend to C4-dominated vegetation as MIS 6 causing a drop in $\delta^{13}C_{org}$ comparable to that of MIS 6. However, MIS 2 is incomplete in the record due to subsequent erosion, in contrast to the more complete section comprising MIS 6. Hence, the comparatively less enriched $\delta^{13}C_{org}$ values from the LGM are likely not fully representative of the vegetation cover changes during this period.

We would also like to point out that the $\delta^{13}C_{org}$ signal is a mixture of organic matter sources, where vegetation shifts (i.e. C3 vs. C4 plants) play a an important role, but also the relative proportion of marine vs. terrestrial constituents. To emphasize the mixed nature of the $\delta^{13}C_{org}$ signal we phrase "... high $\delta^{13}C_{org}$ values as being influenced by enhanced input of POC from C4 plants… ".

D. Do you think that high bottom current itself can result in concentrating the coral pieces like a shell hash deposit, irrespective of high productivity?

It is indeed possible that winnowing due to strong bottom currents leads to the accumulation of larger particles. Winnowing is hence likely responsible for the accumulation of shells hash and coral fragments at the erosive horizons. In general, CWC bearing intervals are characterized by anomalously high bottom current speeds (Fig. 4) and are furthermore embedded in a silt-sized matrix, hence, we do not expect that the vast majority of the CWC-rich intervals have been accumulated by strong bottom current activity.

Please find below the point-by-point responses to the Referees' comments as well as a marked-up manuscript version showing the changes made to the original version of the manuscript.

Please also note that we included Andreas Koutsodendris into the co-author list as he significantly contributed to the revised version of the manuscript.

**Response to Robin Fentimen (Referee 1)**
First we would like to thank Robin Fentimen for his careful and supportive evaluation of our manuscript. Below we provide a point-to-point response (in black) to the original comments (in blue).

Bahr et al. set out to constrain the long-term development of Bowie Mound and to understand the environmental forcing behind its formation. They essentially conclude that an enhanced delivery of terrestrial organic matter during Heinrich Stadials (HS 1, 4 and 6) played an important role on cold-water coral growth off SE Brazil. They compare their set of sedimentological and geochemical proxies to previously published data in the area. As such, the study is based on a solid and plentiful number of proxies. The chronostratigraphy of the core is well constrained and allows, in my opinion, for a satisfying interpretation of the data. The discussion is to the point and not too lengthy, it could even go a bit more in depth (see Comment #3). The stable isotope analyses of infaunal foraminiferal tests is where the most improvement could be done. This is detailed in Specific comment #1. The conclusions drawn by the authors are arguably by the choice of method, and may have yielded more precise results and details if the approach would have been different (and more taxonomically precise; see comment #1).
As detailed below we provide more details on the stable isotope analysis. Although we are aware of the potential limitations of this method (especially of using endobenthic $\delta^{13}$C for bottom water reconstructions due to the absence of sufficient amounts of epibenthic foraminifera) we are confident that the results provide a robust assessment of major shifts in the bottom water regime. However, we agree with the reviewer that an investigation of the benthic foraminiferal communities might provide useful insights into the changes of trophic levels and nutrient supply to the continental margin (such as done in Fentimen et al., 2020). This is clearly beyond the scope of this study but might well be tackled as part of a follow-up research.

The quality of the English in the manuscript is at times insufficient. Some corrections are listed in the section "Technical corrections". In addition to these, the manuscript would need a few extra proof readings to reach the desired quality. I am however confident that this can be done shortly and satisfyingly by the authors.
We thank both Reviewers for pointing at typos and incorrect language. They were all corrected and we also had the text proof-read by a native speaker to improve the quality of the English. Please note that we also changed the title into "Monsoonal forcing of cold-water coral growth off south-eastern Brazil during the past 160 kyrs" instead of "Monsoonal forcing controlled cold water coral growth…".

All things considered, I would be happy to recommend this manuscript for publication in Biogeosciences if the points below (plus the English in the text) are addressed by the authors. The manuscript presents a novel and interesting dataset that falls within the scope of the journal and will be of interest to its readers.
We thank R. Fentimen for his positive assessment of our study and hope that we could sufficiently address the concerns raised in the specific comments below.

Specific comments
Comment #1: My main comment concerns the grouping of different Uvigerina species for stable isotope analyses. The authors mention the genus Uvigerina spp in the material and methods. It would be good to mention here the species considered in this grouping. How many species were considered in the grouping?
We combined *U. peregrina*, and *U. proboscidae*, as now also stated in Section 3.7.1: "For each sample 1–3 tests of *Uvigerina* spp. (*U. peregrina* and *U. proboscidea*)...".

Was one species more abundant? Is one species more abundant during specific intervals (e.g. within CWC bearing intervals)?
The dominant species is *U. peregrina* constituting roughly 75 % of all species of the genus *Uvigerina* in the samples (which rarely contained more than a total of 3 *Uvigerina* specimens). Monospecific selection of *Uvigerina peregrina* would have led to significantly more gaps in the stable isotope record. To obtain enough material we therefore mostly used 3 specimens of *Uvigerina*, when necessary pooling different species for one analysis. We didn't, however, denote the number of individual tests selected from each species per sample. Based on previous inspection of the samples, there is no notable turnover in the relative proportion of the respective species throughout the core, hence, we do not expect a bias in the $\delta^{13}$C record due to shifts in the relative abundances of those species.

Indeed, it has been demonstrated that the response to trophic conditions is species-specific for the genus Uvigerina (see for example, Theodor et al., 2016 Marine Micropaleontology). Uvigerina mediterranea is for example better suited than U. peregrina to reconstruct trophic conditions, since it is more of an opportunistic species. Uvigerinids do not share the same ecological preference (see for example Fontanier et al., 2006), thus I am quite skeptical about this grouping. In my opinion, the grouping of Uvigerinids together weakens the use of stable isotope analyses performed on their tests, since it is not monospecific (as mentioned by the authors at Line 339).

We agree with the reviewer that single-species selection would be the preferred choice for the construction of any stable isotope record, however, we opted for obtaining a record as complete as possible which limited us to combining different species as well as different genera (*Uvigerina* and *Planulina*). As mentioned before, a more in-depth study of faunal turnovers in the benthic foraminiferal communities including stable carbon isotope data from different species with different ecological preferences might be a highly valuable follow-up project but is outside the scope of this study.

We would like to point out that when discussing bottom-water variability as a potential driver of CWC growth we focus on the interpretation of the signal of epibenthic *P. wuellerstorfi* (as stated in the original manuscript text), which provides a more robust insight into bottom-water variability than infaunal species like *Uvigerina*. The higher scatter evident in the *Uvigerina* spp. $\delta^{13}$C data compared to the data obtained on *P. wuellerstorfi* is likely due to the influence of changes in the isotopic composition of pore-water DIC but might also derive from the pooling of different species. However, in the case of *U. peregrina* and *U. proboscidae* Rathburn et al. (1996) found an offset of only 0.1–0.2 ‰. This is much smaller than the 0.8–1.4‰ offset between *U. peregrina* and *U. mediterranea* published by Theodor et al. (2016), indicating that *U. peregrina* and *U. proboscidae* occupy a similar habitat. We nevertheless now state the caveats arising from pooling the different *Uvigerina* species in the manuscript (section 5.1.1):

"The resulting normalized data exhibits a considerable scatter in the *Uvigerina* spp. record due to the pooling of different species, and hence only allows for a discussion of major shifts in the isotopic composition. However, the $\delta^{13}$C data of *Uvigerina* spp. and *P. wuellerstorfi* do not provide compelling evidence for distinctly depleted values during phases of CWC growth compared to CWC-barren intervals (Fig. 4). Although $\delta^{13}C_{Uvi}$ might be influenced by isotopic variations of dissolved inorganic carbon of pore-water (Zahn et al., 1986) and inter-species offsets (Rathburn et al. 1996; Theodor et al., 2016), we nevertheless consider it as appropriate for reconstructing major changes in the bottom-water signature."

Hence the conclusions of section 5.1.1 are not as solid as they could be if authors considered species alone. Although I understand that this approach was chosen as a second choise because of the lack of material, I suggest that the authors should address more and discuss this point more in detail in the material and methods section and in Section 5.1.1.

As suggested, we added further details on the selection of different *Uvigerina* species in the method section and provide a discussion about the caveats in Section 5.1.1 (see comments above).

I recommend plotting the _13C of individual Uvigerina species and then to compare this to the results of the grouping (all species combined). The scatter of the normalized data may possibly be due to the effect of the grouping. This can be easily verified by isolating different Uvigerina species and adding an extra colour code to Fig. 4A. As such, the results presented by the authors would be clearer.

Unfortunately, we did not denote which species we selected for each analysis; in some samples we also had to lump different species of *Uvigerina* into one sample to accumulate enough material. We agree with the reviewer that the grouping might be at least partly responsible for the relatively large scatter of the normalized $\delta^{13}$C data of *Uvigerina* spp.; a respective remark has been added to the discussion in Section 5.1.1 (see above).

As stated above, we nevertheless consider our results obtained from the $\delta^{13}$C analyses to serve its purpose (i.e., tracing bottom-water variability), as we are discussing only major fluctuations in the $\delta^{13}$C signal (not the scatter) and primarily rely on the results of single-species epibenthic *P. wuellerstorfi,* which is naturally a better tracer of bottom-water variability than any endobenthic species.

Comment #2: Although the interpretations and conclusions are in my opinion sound, the association of coral proliferation with HS 4 does not seem as clear as for HS 1 and 6. There is an offset between the Ti/Ca and

speeleotherm records presented in Figure 5 with the coral proliferation phase. Is this due to an age model uncertainty? I think this offset should be discussed a bit more in detail.
We argue that the temporal offsets of CWC proliferation phases *vs.* monsoonal indicators (i.e. the Ti/Ca record and travertine/speleothem growth phases) during HS 2-5 is largely due to age model uncertainties, as the age model of the marine Ti/Ca record is based on AMS 14C dating and benthic isotope stratigraphy, introducing an error of ~2 kyrs for this interval (Campos et al., 2019). It has also to be considered that HS 1 and 6 go along with humid phases that are longer and more pronounced than HS 2-5. Hence, the expected CWC response to the increased run-off should be less pronounced during HS 2-5.

We added these aspects to the discussion of the presumed monsoonal forcing of the CWC signal in Section 5.2: "…The most distinct CWC proliferation phases in fact took place during phases of anomalously strong monsoonal precipitation during the pronounced Heinrich Stadials (HS) 1 and 6 as well as (within age model uncertainties) also during the shorter and less severe humid phases corresponding to HS 2–5 (Fig. 5F, G)."

Comment #3: It would be appreciated if the authors took the discussion one step further by comparing the environmental forcing observed in the study area to other CWC settings, e.g. along the East Atlantic margin or in the Mediterranean. This could be done in the last section of the discussion. For example, Wienberg et al. (2010) suggested that aeolian dust had a local fertilization effect on coral growth in the Gulf of Cadiz, whilst Fentimen et al. (2020) propose that fluvial input triggered coral proliferation during Greenland Interstadial 1 in the Western Mediterranean (Melilla Mound Province). Authors should also consider the work of Mienis et al. in the Western Atlantic.
We value the suggestion to broaden the discussion and now include the reference to other studies that also propose links between terrestrial input and CWC proliferation. These studies include the suggested papers by Wienberg et al. (2010) and Fentimen et al. (2020; Frontiers in Marine Science). We also reference Hanz et al. (2019) who inferred that terrestrial dissolved nutrients aided CWC growth off Angola. We added a respective statement at the end of the first paragraph of Section 5.2: "The here proposed link between Monsoonal activity and CWC growth is in line with studies from the western Mediterranean Sea (Fentimen et al., 2020), the Gulf of Cadiz (Wienberg et al., 2010) and the tropical eastern Atlantic off Angola (Hanz et al., 2019) which inferred that terrestrial input via dust or fluvial run-off ultimately fueled thriving CWC colonies." The mentioned publications of Mienis et al. only referred rather indirectly to terrigenous sediment input as a driver of CWC proliferation, thus we omitted reference to these publications in this context.

As such, the conclusions of the authors fit in with other previous observations and add new evidence. This is something that I believe should be better highlighted and deserves to be developed. The last statement of the conclusion that "This study (. . .) points at a hitherto unrecognized intimate coupling between continental hydroclimate and ecological changes in the deep ocean" is in this sense too bold and should be tempered. Indeed, previous studies already suggest this.
As suggested we toned down the respective sentence to "This study thus presents a prime example of the intimate coupling between continental hydroclimate and ecological changes in the deep ocean." In the same sense we modified the last sentence in the abstract to "Our study thus emphasizes the impact of continental climate variability on a highly vulnerable deep-marine ecosystem".

Also the link between coral growth and monsoonal forcing is only written and stated clearly in the title. No mention of the term "monsoonal forcing" is done in the discussion and conclusion. I think that if the title uses this term, it should also clearly be stated and discussed in the discussion (noticeably in section 5.2).
We now state more explicitly the connection of monsoonal forcing on CWC growth in the discussion (see also response to comment #2) and in the Conclusions ("We find that intervals of high CWC abundance are primarily related to … enhanced monsoonal precipitation in eastern Brazil."). In the abstract we now phrase: "Our results indicate a multi-factorial control on CWC growth and mound formation at Bowie Mound during the past ~160 kyrs, which reveals distinct formation pulses during glacial high northern latitude cold events (Heinrich Stadials, HS) largely associated with anomalously strong monsoonal rainfall over the continent."

Technical corrections Title: "cold-water coral", missing "-"
Corrected.

Line 25: "located at" and not "located in"

Corrected.

 "constrained" and not "constraint"
Corrected.

Lines 48 to 52: These two sentences need to be rephrased; I cannot get the meaning of the sentences as they are. Especially in the second sentence, the verb is missing ("Changes the species (. . .)").
We thank the reviewer for pointing at this typographic mishap. These two sentences now read "The most common framework-forming CWC comprise of *Lophelia pertusa* (recently assigned to the genus *Desmophyllum* by Addamo et al., 2016), *Macropora oculata*, *Solenosmilia variabilis*, *Bathelia candida*, and *Enallopsammia profunda* (e.g., Mangini et al., 2010; Frank et al., 2011; Muñoz et al., 2012; Hebbeln et al., 2014; Raddatz et al., 2020)."

Line 53: Explain the abbreviations POC and DOC the first time you introduce them, some readers may not be acquainted with these.
Done.

Line 55: This sentence needs to be reworked, it is not understandable as it is: "Note, however, that similar studies in the feeding in the properties (. . .)". Do the authors mean feeding properties / feeding behaviour?
This sentence and the following now reads "However, we note that similar studies on the feeding preferences of *S. variabilis*, the dominant framework-building CWC at the herein investigated Bowie Mound (Raddatz et al. 2020) are still missing. Additionally, changes in the properties and spatial configuration of ambient intermediate- or deep-water masses may also strongly impact CWC through changes in the dissolved oxygen concentration and the seawater parameters pH, alkalinity and carbonate-ion concentration."

Line 58: In the sentence: "All affect the capacity (. . .)" I would suggest repeating the word parameters or variables, i.e. "All these parameters (or environmental variables) affect the capacity (. . .)".
The sentence has been revised to "All these parameters affect…".

Line 61: check the grammar: "to play a role in" not "to play a role for"
Corrected.

Lines 62 to 64: The end of this sentence is not clear, consider reworking it. For example: "(. . .) importance of surface productivity in providing food to the deep ocean".
We rephrased the sentence according to the suggestion of the reviewer.

Line 70: I would suggest not to start the sentence with an abbreviation (Here CWC).
We changed the start of the sentence to "The presence of CWC-bearing mounds…".

Line 72: "Adapted" and not "adopted"
Corrected.

Line 82: rephrase the sentence: "demonstrates for the first time" instead of "for the first time demonstrates".
Corrected.

Lines 81 to 83: The combined use in this sentence of "for the first time" and "a so far underestimated" is possibly a bit redundant. I would recommend less emphasizing in this sentence. There is no need to say it is "so far underestimated" if it is the first time it has been observed.
We deleted "for the first time" to avoid redundancy.

Figure 1: Numbers on the hydrographic section (top left) are barely readable. I would suggest increasing the size of these.
Done.

Line 133: Spelling: "half" not "halve"
Corrected.

Line 137: Correct the beginning of the sentence: "Core M125-34-2" instead of "The core (. . .)"
Corrected.

Line 145: Correct the beginning of the sentence: "To constrain" instead of "for constraining"
Corrected.

Line 146: Correct the English: "was sampled at (or sampled every 10 cm)", instead of "was sampled in"
Corrected.

Line 168: "Half" instead of "halve"
Corrected.

Line 181: "at Heidelberg University" instead of "at the Heidelberg University", or rephrase: "at the Department of Geosciences, Heidelberg University".
Corrected.

Line 184: "were analysed with the Diffract Suite (. . .)" instead of "was analysed with Diffract Suite (. . .)"
Corrected.

Line 185: Avoid using the passive form to often when possible. For example here, rather write: "The Rietveld refinement program DIFFRAC.TOPAS (Bruker Software) was used to perform quantitative phase analysis".
Corrected as suggested.

Line 195: "Weighed" instead of "weighted". The verb is "to weigh" (thus weighed in the past tense), the noun is "weight".
Corrected.

Line 195: I would suggest rather writing "filled to the top" instead of "filled until capacity".
Corrected.

Line 198: correct: "(. . .) and put into an ultrasonic bath", instead of "(. . .), put into an (. . .)"
Corrected.

Lines 204 and 205: Is there a mistake here: "The high number of replicates resulted from". Do you mean: "resulted in" ?
We changed the sentence into "The relatively large inter-sample variability that is likely caused by…".

Line 257 and 258: No capital letter given to "core" (write "core")
Corrected.

Figure 3: The symbol (white diamond) of Uvigerina spp. appears to be missing on the figure.
We added the diamond (symbol for *P. wuellerstorfi*) to the legend.

Line 369: correct to "seemed to have"
Corrected.

Line 382 to 384: Check the sentence for grammar: "increasing" instead of "increase", "suggests" instead of "suggest".
We did not apply these suggested changes, as "increasing" does not fit into the sentence's structure and "data", the reference of "suggest" is in plural.

 The sentence needs to be rephrased, it reads: "Due to their baffling capacity, the additional sedimentary input would have aided mound formation". I would recommend rather writing: "Due to the baffling capacity of CWCs, the (. . .)". As it is, the sentence suggests that the mound baffles sediment, whilst it is the corals not the mound in itself.
We thank the reviewer for pointing out this ambiguity. We rephrased the sentence as suggested.

**Response to Referee 2**
First we would like to acknowledge Reviewer 2 for his/her constructive evaluation of our manuscript. Below we provide a point-to-point response (in black) to the original comments (in blue).

The manuscript submitted here investigates the impact of monsoonal variability on CWC growth in last 160 Kyrs. While the authors present a manuscript with compelling arguments; that is likely to be of interest to readers of Biogeosciences, I have a few of concerns that should be addressed before publication.
1. Authors try to show how the monsoon impacted CWC growth without providing any direct correlation between the two, which is a simple statistical analysis to do.
We thank Reviewer 2 for the suggestion of performing a statistical test to validate the proposed monsoonal impact on the CWC growth. For this purpose we computed the linear correlation between CWC occurrences (as shown in Figure 5H) and the Ti/Ca ratio of Core M125-95-3 using the Monte-Carlo-based SurrogateCor function implemented in the "astrochron" package in R (Meyers, 2014), which has been specifically designed for correlation of time series with a different temporal resolution. The resultant correlation coefficient r = 0.56 (p=0.01) corroborates a significant correlation even considering potential mismatches due to age model uncertainties and non-linear proxy behavior. We include these new statistical results in the caption of Fig. 5: "Note the good match between CWC occurrences and enhanced monsoonal activity on the continent (correlation between Ti/Ca and CWC frequency: r = 0.56, p = 0.02; computed using the SurrogateCor function of the R-package "astrochron"; Meyers, 2014)".
Notably, these results also support the already stated results of a discriminant analysis which showed that proxies reflecting terrigenous run-off ($C_{org}/N_{total}$) and weathering (albite/kaolinite) are good predictors for CWC occurrences (cf. end of Section 5.1.3 "Influence of the continental hydrological cycle").

2. The discussion section needs to be streamlined towards the main objective of the manuscript, which now rather seems to be a collection of different points without the central theme. It's difficult for a reader to go through the whole discussion and find exactly where the authors prove their central claim. While discussing many proxies is necessary for a paper like this, it's also important to stress how these proxies help to prove your central claim, which is something lacking in the manuscript.
We recognize the concern by Reviewer 2 (partly also raised by Reviewer 1), and put more emphasis on the main message of the paper, i.e. the direct link between continental hydroclimate and CWC growth at the continental slope. We now stress this link right at the onset of Section 5 "Results and Discussion": "…We argue that the most dominant environmental factor for triggering CWC growth was elevated river run-off during periods of strong monsoonal rainfall in the coastal hinterland, which provided nutrients and organic matter that enhanced the food supply of CWC colonies." We further emphasize the role of monsoonal rainfall as a decisive factor for CWC growth as also requested by Reviewer 1 (this affects Abstract, Section 5.2, and Conclusions).

3. While the growth of CWC during HS events is very evident visually, why the CWC growth was not observed during MIS3 and MIS4 is still not clearly explained. While TOC is the only proxy that was different during these stages but high TOC didn't promote CWC growth at 20-40m [REMARK: this likely refers to 20-40 cm] depth. So it seems that TOC is not a singular factor affecting CWC growth. While authors have explained water currents and terrigenous input as some other proxies to impact CWC growth, they seem to be fluctuating a lot in all the MIS stages and hence fail to shed any light on what stopped CWC from growing during MIS3 and MIS4.
The Reviewer refers here to the CWC barren interval between 20–40 cm core depth. We note that this interval is characterized by intermediate TOC contents that are not as high as during the main CWC growth phases between 70-180 cm and below 530 c (Fig. 6A). As already stated in the original text (see section 5.1.2: "However, as suggested in the previous section, there are underline{multiple factors necessary} for stimulating coral growth at Bowie Mound"), we assume a multi-factorial control of CWC proliferation phases. Organic matter supply likely played a major role but other factors such as hydrodynamic conditions might have interfered as well.
In this line the absence of CWC during large portions of MIS 4 could be well explained by the generally low TOC contents that point to the lack of organic matter supply inhibiting CWC growth. We now state this connection explicitly in Section 5.1.2: "…while the long CWC-barren interval between 200–460 cm is characterized by relatively low TOC contents…". According to the stratigraphic assignment this specific

interval encompasses early MIS 4, not MIS 3; to make this distinction clearer we modified the age assignments in Figures 4 and 6 and modified the respective paragraph in Section 4:

"The section between $E_{II}$ and $E_{III}$ on the other hand has relatively uniform δ$^{18}$O values around 4.3 ‰ (δ$^{18}$O$_{Uvi}$) and 3.4 ‰ (δ$^{18}$O$_{Plan}$), respectively, which matches M125-50-3 δ$^{18}$O$_{Uvi}$ values during MIS 4 to 2. A Th/U date at the top of this section at 117 cm reveals an age of 34 ka, while CWC ages from slightly deeper in the core (between 131-190 cm) fall within the range of 60–63 ka (MIS 4). As δ$^{18}$O$_{Uvi}$ values between $E_{II}$ and $E_{III}$ are less depleted than MIS 5 samples of reference site M125-50-3, we infer that those sediments were most likely deposited during MIS 4 and did not reach into MIS 5. Hence, it hence appears that deposits of MIS 2 and large parts of MIS 3 are not present in core M125-34-2, either due to non-deposition or subsequent erosion (note the prominent erosive surface $E_{II}$). This age assignment would also imply that the extended CWC-free portion from 200 to 465 cm was deposited within a short period of approximately 8 kyr during MIS 4 (62.2 ka as the oldest Th/U dates and ~70 ka as the MIS 4/5 boundary)."

4. The figures captions throughout the manuscript describe what is shown in the figure, but don't provide the reader with any additional information such as calling attention to the significant result. The message shown by the figure is left entirely up to the reader to decipher.

We are aware that the figures and the associated discussion of the factors influencing CWC growth phases are quite complex. We therefore acknowledge the suggestion by Reviewer 2 to provide more information on the interpretation of a Figure's content in the respective caption. We hence added the following sentences:

Fig. 4: "Phases of CWC proliferation appear to require background state of high hydrodynamics conditions (elevated ln(Zr/Al) and $\overline{SS}$) but do not show an influence of deep-water mass variability ($\square^{13}$C).."

Fig. 5: "Note the good match between CWC occurrences and enhanced monsoonal activity on the continent (correlation between ln(Ti/Ca) and CWC frequency: r = 0.56, p = 0.02; computed using the SurrogateCor function of the R-package "astrochron"; Meyers, 2014)."

Fig. 6: "Note that high CWC abundances fall into intervals of high TOC and increased weathering due to an intensified continental hydrological cycle."

Moreover, in some figures authors have added depth and in some age. It would be best if authors add age and depth in all the figures.

We would like to note that we already provide depth and age scales on Figures, when appropriate (Figs. 3, 4, 6). For Figure 5 we followed the reviewer's suggestion and added the depths intervals denoting the phases of CWC proliferation at Bowie Mound on top of the figure; a respective comment has been added to the caption ("Red bars indicate periods of enhanced CWC growth at Bowie Mound, with the respective depths in Core M125-34-2 annotated").

5. Line 48: "The most common framework-forming CWC comprise. "This sentence doesn't make sense. It is either incomplete or needs to be restructured.

This sentence has been rephrased (see also reply to respective comment by Reviewer 1).

6. The next sentence in line 49 "Changes the species. . .." Is also incomplete and hence doesn't provide context.

This sentence has been rephrased (see also reply to respective comment by Reviewer 1).

7. Line 55: "in" repeated "similar studies in the feeding in the properties. . ."

We rephrased the sentence to "… similar studies on the feeding preferences of *S. variabilis*, the dominant framework-building CWC discussed in this study, are still pending".

8. Line 72: It should be "adapted" instead of "adopted".

Corrected.

9. Line 77 -79: "This setting allows us. . ... growth at Bowie mound". It is a repetition of what has been already said in previous sentences.

This sentence has been removed (see also reply to respective comment by Reviewer 1).

10. Line 369: "AAIW seemed to had an insignificant" It should be "to have had" or "AAIW had" depending on what authors want to say exactly.
Corrected to "seemed to have".

11. Line 384: "does not necessarily led to". It should be "lead"
Corrected.

[revised manuscript text omitted]

Segment 1, 0–83 cm

X-Ray slice 20 — Core photo

13.7±0.1 ka
14.3±0.1 ka
16.5±0.1 ka

erosional contact ● Th/U dating

Segment 1, 0–83 cm

X-Ray slice 20 — Core photo

13.7±0.1 ka
14.3±0.1 ka

[revised manuscript text omitted]

Segment 1, 0–83 cm

[Figure]

X-Ray
slice 20

Core
photo

Segment 1, 0–83 cm

X-Ray
slice 20

Core
photo

13.7±0.1 ka

14.3±0.1 ka

16.5±0.1 ka

13.7±0.1 ka

14.3±0.1 ka

16.5±0.1 ka

erosional contact    Th/U dating

erosional contact    Th/U dating

[revised manuscript text omitted]